# FilterNet: Harnessing Frequency Filters for Time Series Forecasting

**Kun Yi**[1,2]**, Jingru Fei**[3]**, Qi Zhang**[4]**, Hui He**[3]**, Shufeng Hao**[5]**, Defu Lian**[6]**, Wei Fan**[7][*]

[1]North China Institute of Computing Technology, [2]State Information Center of China
[3]Beijing Institute of Technology, [4]Tongji University, [5]Taiyuan University of Technology
[6]University of Science and Technology of China, [7]University of Oxford
kunyi.cn@gmail.com, {jingrufei, hehui617}@bit.edu.cn, zhangqi_cs@tongji.edu.cn
haoshufeng@tyut.edu.cn, liandefu@ustc.edu.cn, weifan.oxford@gmail.com

## Abstract

Given the ubiquitous presence of time series data across various domains, precise forecasting of time series holds significant importance and finds widespread real-world applications such as energy, weather, healthcare, etc. While numerous forecasters have been proposed using different network architectures, the Transformer-based models have state-of-the-art performance in time series forecasting. However, forecasters based on Transformers are still suffering from vulnerability to high-frequency signals, efficiency in computation, and bottleneck in full-spectrum utilization, which essentially are the cornerstones for accurately predicting time series with thousands of points. In this paper, we explore a novel perspective of enlightening *signal processing* for deep time series forecasting. Inspired by the *filtering* process, we introduce one simple yet effective network, namely *FilterNet*, built upon our proposed learnable *frequency filters* to extract key informative temporal patterns by selectively passing or attenuating certain components of time series signals. Concretely, we propose two kinds of learnable filters in the FilterNet: (i) Plain shaping filter, that adopts a universal frequency kernel for signal filtering and temporal modeling; (ii) Contextual shaping filter, that utilizes filtered frequencies examined in terms of its compatibility with input signals for dependency learning. Equipped with the two filters, FilterNet can approximately surrogate the linear and attention mappings widely adopted in time series literature, while enjoying superb abilities in handling high-frequency noises and utilizing the whole frequency spectrum that is beneficial for forecasting. Finally, we conduct extensive experiments on eight time series forecasting benchmarks, and experimental results have demonstrated our superior performance in terms of both effectiveness and efficiency compared with state-of-the-art methods. Our code is available at [1].

## 1 Introduction

Time series forecasting has been playing a pivotal role across a multitude of contemporary applications, spanning diverse domains such as climate analysis [1], energy production [2], traffic flow patterns [3], financial markets [4], and various industrial systems [5]. The ubiquity and profound significance of time series data has recently garnered substantial research efforts, culminating in a plethora of deep learning forecasting models [6] that have significantly enhanced the domain of time series forecasting.

Previously, leveraging different kinds of deep neural networks derives a series of time series forecasting methods, such as Recurrent Neural Network-based methods including DeepAR [7], LSTNet [8],

---

[*]Corresponding author
[1]https://github.com/aikunyi/FilterNet

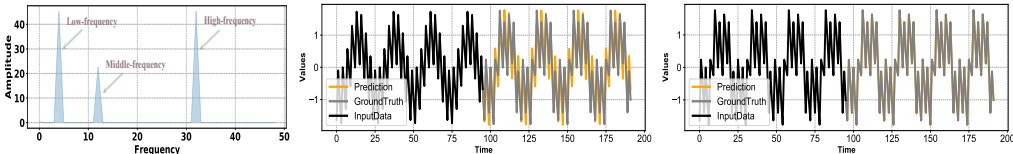

(a) The spectrum of input signal (b) iTransformer with MSE=1.1e-01 (c) FilterNet with MSE=2.7e-05

Figure 1: Performance of Mean Squared Error (MSE) on a simple synthetic multi-frequency signal. More details about the experimental settings can be found in Appendix C.4.

Convolution Neural Network-based methods including TCN [9], SCINet [10], etc. Recently, however, with the continuing advancement of deep learning, two branches of methods that received particularly more attention have been steering the development of time series forecasting, i.e., Multilayer Perceptron (MLP)-based methods, such as N-BEATS [11], DLinear [12], and FreTS [13], and Transformer-based methods, such as Informer [14], Autoformer [15], PatchTST [16], and iTransformer [17]. While MLP-based models are capable of providing accurate forecasts, Transformer-based models continue to achieve state-of-the-art time series forecasting performance.

However, forecasters based on Transformers are still suffering from *vulnerability* to high-frequency signals, *efficiency* in computation, and *bottleneck* in full-spectrum utilization, which essentially are the cornerstones for accurately predicting time series composed of thousands of timesteps. In designing a very simple simulation experiment on the synthetic data only composed of a low-, middle- and high-frequency signal respectively (see Figure 1(a)), we find the state-of-the-art iTransformer [17] model performs much worse in forecasting (Figure 1(b) and Figure 1(c)). This observation shows that state-of-the-art Transformer-based model cannot utilize the full spectrum information, even for a naive signal of three different frequency components. In contrast, in the field of *signal processing*, a *frequency filter* enjoys many good properties such as frequency selectivity, signal conditioning, and multi-rate processing. These could have great potential in advancing the model's ability to extract key informative frequency patterns in time series forecasting.

Thus, inspired by the *filtering* process [18] in signal processing, in this paper, we introduce one simple yet effective framework, namely *FilterNet*, for effective time series forecasting. Specifically, we start by proposing two kinds of learnable filters as the key units in our framework: (i) Plain shaping filter, which makes the naive but universal frequency kernel learnable for signal filtering and temporal modeling, and (ii) Contextual shaping filter, which utilizes filtered frequencies examined in terms of its compatibility with input signals for dependency learning. The plain shaping filter is more likely to be adopted in predefined conditions and efficient in handling simple time series structures, while the contextual filter can adaptively weight the filtering process based on the changing conditions of input and thus have more flexibility in facing more complex situations. Besides, these two filters as the built-in functions of the FilterNet can also approximately surrogate the widely adopted linear mappings and attention mappings in time series literature [12, 14, 17]. This also illustrates the effectiveness of our FilterNet in forecasting by selectively passing or attenuating certain signal components while capturing the core time series structure with adequate learning expressiveness. Moreover, since filters are better fit in the stationary frequency domain, we let filters wrapped by two reversible transformations, i.e., instance normalization [19] and fast Fourier transform [20] to reduce the influence of non-stationarity and accomplish the domain transformation of time series respectively. In summary, our contributions can be listed as follows:

- In studying state-of-the-art deep Transformer-based time series forecasting models, an interesting observation from a simple simulation experiment motivates us to explore a novel perspective of enlightening *signal processing* techniques for deep time series forecasting.

- Inspired by the *filtering* process in signal processing, we introduce a simple yet effective network, *FilterNet*, built upon our proposed two learnable frequency filters to extract key informative temporal patterns by selectively passing or attenuating certain components of time series signals, thereby enhancing the forecasting performance.

- We conduct extensive experiments on eight time series forecasting benchmarks, and the results have demonstrated that our model achieves superior performance compared with state-of-the-art forecasting algorithms in terms of effectiveness and efficiency.

## 2 Related Work

**Deep Learning-based Time Series Forecasting**  In recent years, deep learning-based methods have gained prominence in time series forecasting due to their ability to capture nonlinear and complex correlations [21]. These methods have employed various network architectures to learn temporal dependencies, such as RNN [8, 7], TCN [9, 10], etc. Notably, MLP- and Transformer-based methods have achieved competitive performance, emerging as state-of-the-art approaches. N-HiTS [22] integrates multi-rate input sampling and hierarchical interpolation with MLPs to enhance univariate forecasting. DLinear [12] introduces a simple approach using a single-layer linear model to capture temporal relationships between input and output time series data. RLinear [23] utilizes linear mapping to model periodic features, demonstrating robustness across diverse periods with increasing input length. In contrast to the simple structure of MLPs, Transformer's advanced attention mechanisms [24] empower the models [14, 15, 25, 26] to capture intricate dependencies and long-range interactions. PatchTST [16] segments time series into patches as input tokens to the Transformer and maintaining channel independence. iTransformer [17] inverts the Transformer's structure by treating independent series as variate tokens to capture multivariate correlations through attention.

**Time Series Modeling with Frequency Learning**  In recent developments, frequency technology has been increasingly integrated into deep learning models, significantly improving state-of-the-art accuracy and efficiency in time series analysis [27]. These models leverage the benefits of frequency technology, such as high efficiency [28, 29] and energy compaction [13], to enhance forecasting capabilities. Concretely, Autoformer [15] introduces the auto-correlation mechanism, improving self-attention implemented with Fast Fourier Transforms (FFT). FEDformer [25] enhances attention with a FFT-based frequency approach, determining attentive weights from query and key spectrums and conducting weighted summation in the frequency domain. DEPTS [30] utilizes Discrete Fourier Transform (DFT) to extract periodic patterns and contribute them to forecasting. FiLM [31] employs Fourier analysis to retain historical information while filtering out noisy signals. FreTS [13] introduces a frequency-domain Multi-Layer Perceptrons (MLPs) to learn channel and temporal dependencies. FourierGNN [29] transfers the operations of graph neural networks (GNNs) from the time domain to the frequency domain. FITS [32] applies a low pass filter to the input data followed by complex-valued linear mapping in the frequency domain.

Unlike these methods, in this paper we propose a simple yet effective model FilterNet developed from a signal processing perspective, and apply all-pass frequency filters to design the network directly, rather than incorporating them into other network architectures, such as Transformers, MLPs, or GNNs, or utilizing them as low-pass filters, as done in FITS [32] and FiLM [31].

## 3 Preliminaries

**Frequency Filters**  Frequency filters [33] are mathematical operators designed to modify the spectral content of signals. Specifically, given an input time series signal $x[n]$ with its corresponding Fourier transform $\mathcal{X}[k]$, a frequency filter $\mathcal{H}[k]$ is applied to the signal to produce an output signal $y[n]$ with its corresponding Fourier transform $\mathcal{Y}[k] = \mathcal{X}[k]\mathcal{H}[k]$. The frequency filter $\mathcal{H}[k]$ alters the amplitude and phase of specific frequency components in the input time series signal $x[n]$ according to its *frequency response characteristics*, thereby shaping the spectral content of the output signal.

According to the Convolution Theorem [34], the point-wise multiplication in the frequency domain corresponds to the circular convolution operation between two corresponding signals in the time domain. Consequently, the frequency filter process can be expressed in the time domain as:

$$\mathcal{Y}[k] = \mathcal{H}[k]\mathcal{X}[k] \leftrightarrow y[n] = h[n] \circledast x[n], \tag{1}$$

where $h[n]$ is the inverse Fourier transform of $\mathcal{H}[k]$ and $\circledast$ denotes the circular convolution operation. This formulation underscores the intrinsic connections between the frequency filter process and the circular convolution in the time domain, and it indicates that the frequency filter process can efficiently perform circular convolution operations. In time series forecasting, Transformer-based methods have achieved state-of-the-art performance, largely due to the self-attention mechanism [14, 15, 25, 17], which can be interpreted as a form of global circular convolution [35]. This perspective opens up the possibility of integrating frequency filter technologies, which are well-known for their ability to isolate and enhance specific signal components, to further improve time series forecasting models.

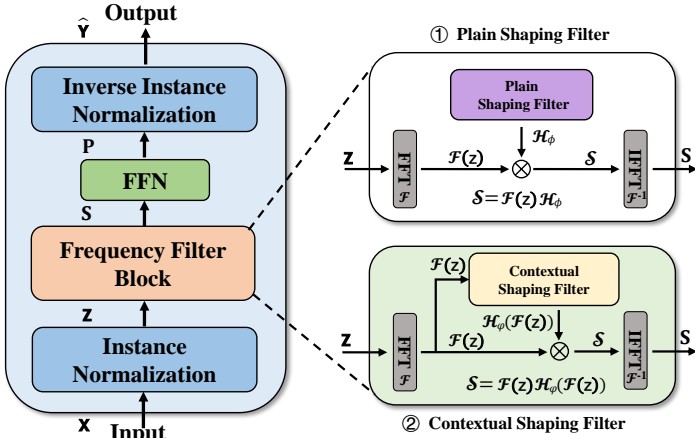

Figure 2: The overall architecture of FilterNet. (i) Instance normalization is employed to address the non-stationarity among time series data; (ii) The frequency filter block is applied to capture the temporal patterns, which has two different implementations, i.e., plain shaping filter and contextual shaping filter; (iii) Feed-forward network is adopted to project the temporal patterns extracted by frequency filter block back onto the time series data and make predictions.

## 4 Methodology

As aforementioned, frequency filters enjoy numerous advantageous properties for time series forecasting, functioning equivalently to circular convolution operations in the time domain. Therefore, we design the time series forecaster from the perspective of frequency filters. In this regard, we propose *FilterNet*, a forecasting architecture grounded in frequency filters. First, we introduce the overall architecture of FilterNet in Section 4.1, which primarily comprises the basic blocks and the frequency filter block. Second, we delve into the details of two types of frequency filter blocks: the *plain shaping filter* presented in Section 4.2 and the *contextual shaping filter* discussed in Section 4.3.

### 4.1 Overview

The overall architecture of FilterNet is depicted in Figure 2, which mainly consists of the instance normalization part, the frequency filter block, and the feed-forward network. Specifically, for a given time series input $\mathbf{X} = [X_1^{1:L}, X_2^{1:L}, ..., X_N^{1:L}] \in \mathbb{R}^{N \times L}$ with the number of variables $N$ and the lookback window length $L$, where $X_N^{1:L} \in \mathbb{R}^L$ denotes the $N$-th variable, we employ FilterNet to predict the future $\tau$ time steps $\mathbf{Y} = [X_1^{L+1:L+\tau}, X_2^{L+1:L+\tau}, ..., X_N^{L+1:L+\tau}] \in \mathbb{R}^{N \times \tau}$. We provide further analysis about the architecture design of FilterNet in Appendix A.

**Instance Normalization** Non-stationarity is widely existing in time series data and poses a crucial challenge for accurate forecasting [19, 36]. Considering that time series data are typically collected over a long duration, these non-stationary sequences inevitably expose forecasting models to distribution shifts over time. Such shifts can result in performance degradation during testing due to the covariate shift or the conditional shift [37]. To address this problem, we utilize an instance normalization method, denoted as Norm, on the time series input $\mathbf{X}$, which can be formulated as:

$$\text{Norm}(\mathbf{X}) = [\frac{X_i^{1:L} - \text{Mean}_L(X_i^{1:L})}{\text{Std}_L(X_i^{1:L})}]_{i=1}^N, \tag{2}$$

where $\text{Mean}_L$ denotes the operation that calculates the mean value along the time dimension, and $\text{Std}_L$ represents the operation that calculates the standard deviation along the time dimension.

Correspondingly, the inverse instance normalization, denoted as InverseNorm, is formulated as:

$$\text{InverseNorm}(\mathbf{P}) = [P_i^{L+1:L+\tau} \times \text{Std}_L(X_i^{1:L}) + \text{Mean}_L(X_i^{1:L})]_{i=1}^N, \tag{3}$$

where $\mathbf{P} = [P_1^{L+1:L+\tau}, P_2^{L+1:L+\tau}, ..., P_N^{L+1:L+\tau}] \in \mathbb{R}^{N \times \tau}$ are the predictive values.

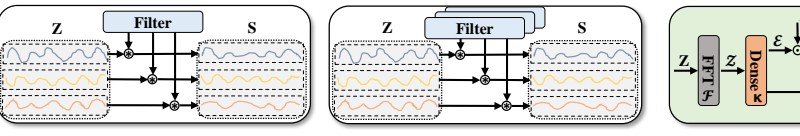

(a) Plain Shaping Filter          (b) Contextual Shaping Filter

Figure 3: The structure of frequency filters. (a) Plain shaping filter: the plain shaping filter is initialized randomly with channel-shared (left) or channel-unique (right) parameters, and then performs circular convolution (i.e., the symbol $\circledast$) with the input time series; (b) Contextual shaping filter: the contextual shaping filter firstly learns a data-dependent filter and then conducts multiplication (i.e., the symbol $\odot$) with the frequency representation of the input time series.

**Frequency Filter Block**  Previous representative works primarily leverage MLP architectures (e.g., DLinear [12], RLinear [23]) or Transformer architectures (e.g., PatchTST [16], iTransformer [17]) to model the temporal dependencies among time series data. As mentioned earlier, time series forecasters can be implemented through performing a *frequency filter process* in the frequency domain, and thus we propose to directly apply the frequency filter in the frequency domain, denoted as FilterBlock, to replace the aforementioned methods for modeling corresponding temporal dependencies, such as:

$$\text{FilterBlock}(\mathbf{Z}) = \mathcal{F}^{-1}(\mathcal{F}(\mathbf{Z})\mathcal{H}_{filter}), \tag{4}$$

where $\mathcal{F}$ is Fourier transform, $\mathcal{F}^{-1}$ is inverse Fourier transform and $\mathcal{H}_{filter}$ is the frequency filter.

Inspired by MLP that randomly initializes a learnable weight parameters and Transformer that learns the data-dependent attention scores from data (further explanations are provided in Appendix B), we introduce two types of frequency filters, i.e., ***plain shaping filter*** (PaiFilter) and *contextual shaping **filter*** (TexFilter). PaiFilter applies a random initialized learnable weight $\mathcal{H}_{\phi}$ to instantiate the frequency filter $\mathcal{H}_{filter}$, and then the frequency filter process is reformulated as:

$$\text{FilterBlock}(\mathbf{Z}) = \mathcal{F}^{-1}(\mathcal{F}(\mathbf{Z})\mathcal{H}_{\phi}). \tag{5}$$

TexFilter learns a data-dependent frequency filter $\mathcal{H}_{\varphi}(\mathcal{F}(\mathbf{Z}))$ from the input data by using a neural network $\mathcal{H}_{\varphi}()$, and then the corresponding frequency filter process is reformulated as:

$$\text{FilterBlock}(\mathbf{Z}) = \mathcal{F}^{-1}(\mathcal{F}(\mathbf{Z})\mathcal{H}_{\varphi}(\mathcal{F}(\mathbf{Z}))). \tag{6}$$

**Feed-forward Network**  The frequency filter block has captured temporal dependencies among time series data, and then we employ a feed-forward network (FFN) to project them back onto the time series data and make predictions for the future $\tau$ time steps. As the output $\mathbf{P}$ of FFN are instance-normalized values, we conduct an inverse instance normalization operation (InverseNorm) on them and obtain the final predictions $\hat{\mathbf{Y}}$. The entire process can be formulated as follows:

$$\mathbf{P} = \text{FFN}(\mathbf{S}), \tag{7}$$
$$\hat{\mathbf{Y}} = \text{InverseNorm}(\mathbf{P}).$$

## 4.2  Plain Shaping Filter

PaiFilter instantiates the frequency filter by randomly initializing learnable parameters and then performing multiplication with the input time series. In general, for multivariate time series data, the channel-independence strategy in channel modeling has proven to be more effective compared to the channel-mixing strategy [12, 16]. Following this principle, we also adopt the channel-independence strategy for designing the frequency filter. Specifically, we propose two types of plain shaping filters: the universal type, where parameters are shared across different channels, and the individual type, where parameters are unique to each channel, as illustrated in Figure 3(a).

Given the time series input $\mathbf{Z} \in \mathbb{R}^{N \times L}$ and the plain shaping filter $\mathcal{H}_{\phi}$, we apply PaiFilter by:

$$\begin{aligned} \mathcal{Z} &= \mathcal{F}(\mathbf{Z}), \\ \mathcal{S} &= \mathcal{Z} \odot_L \mathcal{H}_{\phi}, \quad \mathcal{H}_{\phi} \in \{\mathcal{H}_{\phi}^{(Uni)}, \mathcal{H}_{\phi}^{(Ind)}\} \\ \mathbf{S} &= \mathcal{F}^{-1}(\mathcal{S}), \end{aligned} \tag{8}$$

where $\mathcal{F}$ is Fourier transform, $\mathcal{F}^{-1}$ is inverse Fourier transform, $\odot_L$ denotes the element-wise product along $L$ dimension, $\mathcal{H}_\phi^{(Uni)} \in \mathbb{C}^{1 \times L}$ is the universal plain shaping filter, $\mathcal{H}_\phi^{(Ind)} \in \mathbb{C}^{N \times L}$ is the individual plain shaping filter, and $\mathbf{S} \in \mathbb{R}^{N \times L}$ is the output of PaiFilter. We further compare and analyze the two types of PaiFilter in Section 5.3.

### 4.3 Contextual Shaping Filter

In contrast to PaiFilter, which randomly initializes the parameters of frequency filters and fixes them after training, TexFilter learns the parameters generated from the input data, allowing for better adaptation to the data. Consequently, we devise a neural network $\mathcal{H}_\varphi$ that flexibly adjusts the frequency filter in response to the input data, as depicted in Figure 3(b).

Given the time series input $\mathbf{Z} \in \mathbb{R}^{N \times L}$ and its corresponding Fourier transform denoted as $\mathcal{Z} = \mathcal{F}(\mathbf{Z}) \in \mathbb{C}^{N \times L}$, the network $\mathcal{H}_\varphi$ is utilized to derive the contextual shaping filter, expressed as $\mathcal{H}_\varphi : \mathbb{C}^{N \times L} \mapsto \mathbb{C}^{N \times D}$. First, it embeds the raw data by a linear dense operation $\kappa : \mathbb{C}^L \mapsto \mathbb{C}^D$ to improve the capability of modeling complex data. Then, it applies a series of complex-value multiplication with $K$ learnable parameters $\mathcal{W}_{1:K} \in \mathbb{C}^{1 \times D}$ yielding $\sigma(\kappa(\mathcal{Z}) \odot \mathcal{W}_{1:K})$ where $\sigma$ is the activation function, and finally outputs $\mathcal{H}_\varphi(\mathcal{Z})$. Then we apply TexFilter by:

$$
\begin{aligned}
\mathcal{Z} &= \mathcal{F}(\mathbf{Z}), \\
\mathcal{E} &= \kappa(\mathcal{Z}), \\
\mathcal{H}_\varphi(\mathcal{Z}) &= \sigma(\mathcal{E} \odot_D \mathcal{W}_{1:K}), \quad \mathcal{W}_{1:K} = \prod_{i=1}^{K} \mathcal{W}_i \\
\mathcal{S} &= \mathcal{E} \odot_D \mathcal{H}_\varphi(\mathcal{Z}), \\
\mathbf{S} &= \mathcal{F}^{-1}(\mathcal{S}),
\end{aligned}
\tag{9}
$$

where $\odot_D$ denotes the element-wise product along $D$ dimension and $\mathbf{S} \in \mathbb{R}^{N \times D}$ is the output. The contextual shaping filter can adaptively weight the filtering process based on the changing conditions of input and thus have more flexibility in facing more complex situations.

## 5 Experiments

In this section, we extensively experiment with eight real-world time series benchmarks to assess the performance of our proposed FilterNet. Furthermore, we conduct thorough analytical experiments concerning the frequency filters to validate the effectiveness of our proposed framework.

### 5.1 Experimental Setup

**Datasets** We conduct empirical analyses on diverse datasets spanning multiple domains, including traffic, energy, and weather, among others. Specifically, we utilize datasets such as ETT datasets [14], Exchange [38], Traffic [15], Electricity [15], and Weather [15], consistent with prior studies on long time series forecasting [16, 17, 25]. We preprocess all datasets according to the methods outlined in [16, 17], and normalize them with the standard normalization method. We split the datasets into training, validation, and test sets in a 7:2:1 ratio. More dataset details are in Appendix C.1.

**Baselines** We compare our proposed FilterNet with the representative and state-of-the-art models to evaluate their effectiveness for time series forecasting. We choose the baseline methods from four categories: (1) Frequency-based models, including FreTS [13] and FITS [32]; (2) TCN-based models, such as MICN [39] and TimesNet [40]; (3) MLP-based models, namely DLinear [12] and RLinear [23]; and (4) Transformer-based models, which include Informer [14], Autoformer [15], Pyraformer [26], FEDformer [25], PatchTST [16], and the more recent iTransformer [17] for comparison. Further details about the baselines can be found in Appendix C.2.

**Implementation Details** All experiments are implemented using Pytorch 1.8 [41] and conducted on a single NVIDIA RTX 3080 10GB GPU. We employ MSE (Mean Squared Error) as the loss function and present MAE (Mean Absolute Errors) and MSE (Mean Squared Errors) results as the evaluation metrics. For further implementation details, please refer to Appendix C.3.

Table 1: Forecasting results for prediction lengths $\tau \in \{96, 192, 336, 720\}$ with lookback window length $L = 96$. The best results are in red and the second best are blue. Due to space limit, additional results with other baselines and under different lookback length are provided in Tables 4 and 5.

| Models | | TexFilter | | PaiFilter | | iTransformer | | PatchTST | | FEDformer | | TimesNet | | DLinear | | RLinear | | FITS | |
|---|---|---|---|---|---|---|---|---|---|---|---|---|---|---|---|---|---|---|---|
| Metrics | | MSE | MAE | MSE | MAE | MSE | MAE | MSE | MAE | MSE | MAE | MSE | MAE | MSE | MAE | MSE | MAE | MSE | MAE |
| ETTm1 | 96 | 0.321 | 0.361 | 0.318 | 0.358 | 0.334 | 0.368 | 0.329 | 0.367 | 0.379 | 0.419 | 0.338 | 0.375 | 0.344 | 0.370 | 0.355 | 0.376 | 0.355 | 0.375 |
| | 192 | 0.367 | 0.387 | 0.364 | 0.383 | 0.377 | 0.391 | 0.367 | 0.385 | 0.426 | 0.441 | 0.374 | 0.387 | 0.379 | 0.393 | 0.387 | 0.392 | 0.392 | 0.393 |
| | 336 | 0.401 | 0.409 | 0.396 | 0.406 | 0.426 | 0.420 | 0.399 | 0.410 | 0.445 | 0.459 | 0.410 | 0.411 | 0.410 | 0.411 | 0.424 | 0.415 | 0.424 | 0.414 |
| | 720 | 0.477 | 0.448 | 0.456 | 0.444 | 0.491 | 0.459 | 0.454 | 0.439 | 0.543 | 0.490 | 0.478 | 0.450 | 0.473 | 0.450 | 0.487 | 0.450 | 0.487 | 0.449 |
| ETTm2 | 96 | 0.175 | 0.258 | 0.174 | 0.257 | 0.180 | 0.264 | 0.175 | 0.259 | 0.203 | 0.287 | 0.187 | 0.267 | 0.187 | 0.281 | 0.182 | 0.265 | 0.183 | 0.266 |
| | 192 | 0.240 | 0.301 | 0.240 | 0.300 | 0.250 | 0.309 | 0.241 | 0.302 | 0.269 | 0.328 | 0.249 | 0.309 | 0.272 | 0.349 | 0.246 | 0.304 | 0.247 | 0.305 |
| | 336 | 0.311 | 0.347 | 0.297 | 0.339 | 0.311 | 0.348 | 0.305 | 0.343 | 0.325 | 0.366 | 0.321 | 0.351 | 0.316 | 0.372 | 0.307 | 0.342 | 0.307 | 0.342 |
| | 720 | 0.414 | 0.405 | 0.392 | 0.393 | 0.412 | 0.407 | 0.402 | 0.400 | 0.421 | 0.415 | 0.408 | 0.403 | 0.452 | 0.457 | 0.407 | 0.398 | 0.407 | 0.399 |
| ETTh1 | 96 | 0.382 | 0.402 | 0.375 | 0.394 | 0.386 | 0.405 | 0.414 | 0.419 | 0.376 | 0.420 | 0.384 | 0.402 | 0.383 | 0.396 | 0.386 | 0.395 | 0.386 | 0.396 |
| | 192 | 0.430 | 0.429 | 0.436 | 0.422 | 0.441 | 0.436 | 0.460 | 0.445 | 0.420 | 0.448 | 0.436 | 0.429 | 0.433 | 0.426 | 0.437 | 0.424 | 0.436 | 0.423 |
| | 336 | 0.472 | 0.451 | 0.476 | 0.443 | 0.487 | 0.458 | 0.501 | 0.466 | 0.459 | 0.465 | 0.491 | 0.469 | 0.479 | 0.457 | 0.479 | 0.446 | 0.478 | 0.444 |
| | 720 | 0.481 | 0.473 | 0.474 | 0.469 | 0.503 | 0.491 | 0.500 | 0.488 | 0.506 | 0.507 | 0.521 | 0.500 | 0.517 | 0.513 | 0.481 | 0.470 | 0.502 | 0.495 |
| ETTh2 | 96 | 0.293 | 0.343 | 0.292 | 0.343 | 0.297 | 0.349 | 0.302 | 0.348 | 0.358 | 0.397 | 0.340 | 0.374 | 0.320 | 0.374 | 0.318 | 0.363 | 0.295 | 0.350 |
| | 192 | 0.374 | 0.396 | 0.369 | 0.395 | 0.380 | 0.400 | 0.388 | 0.400 | 0.429 | 0.439 | 0.402 | 0.414 | 0.449 | 0.454 | 0.401 | 0.412 | 0.381 | 0.396 |
| | 336 | 0.417 | 0.430 | 0.420 | 0.432 | 0.428 | 0.432 | 0.426 | 0.433 | 0.496 | 0.487 | 0.452 | 0.452 | 0.467 | 0.469 | 0.436 | 0.442 | 0.426 | 0.438 |
| | 720 | 0.449 | 0.460 | 0.430 | 0.446 | 0.427 | 0.445 | 0.431 | 0.446 | 0.463 | 0474 | 0.462 | 0.468 | 0.656 | 0.571 | 0.442 | 0.454 | 0.431 | 0.446 |
| ECL | 96 | 0.147 | 0.245 | 0.176 | 0.264 | 0.148 | 0.240 | 0.181 | 0.270 | 0.193 | 0.308 | 0.168 | 0.272 | 0.195 | 0.277 | 0.201 | 0.281 | 0.200 | 0.278 |
| | 192 | 0.160 | 0.250 | 0.185 | 0.270 | 0.162 | 0.253 | 0.188 | 0.274 | 0.201 | 0.315 | 0.184 | 0.289 | 0.194 | 0.280 | 0.201 | 0.283 | 0.200 | 0.280 |
| | 336 | 0.173 | 0.267 | 0.202 | 0.286 | 0.178 | 0.269 | 0.204 | 0.293 | 0.214 | 0.329 | 0.198 | 0.300 | 0.207 | 0.296 | 0.215 | 0.298 | 0.214 | 0.295 |
| | 720 | 0.210 | 0.309 | 0.242 | 0.319 | 0.225 | 0.317 | 0.246 | 0.324 | 0.246 | 0.355 | 0.220 | 0.320 | 0.242 | 0.329 | 0.257 | 0.331 | 0.255 | 0.327 |
| Exchange | 96 | 0.091 | 0.211 | 0.083 | 0.202 | 0.086 | 0.206 | 0.088 | 0.205 | 0.148 | 0.278 | 0.107 | 0.234 | 0.085 | 0.210 | 0.093 | 0.217 | 0.084 | 0.203 |
| | 192 | 0.186 | 0.305 | 0.174 | 0.296 | 0.177 | 0.299 | 0.176 | 0.299 | 0.271 | 0.315 | 0.226 | 0.344 | 0.178 | 0.299 | 0.184 | 0.307 | 0.177 | 0.298 |
| | 336 | 0.380 | 0.449 | 0.326 | 0.413 | 0.331 | 0.417 | 0.301 | 0.397 | 0.460 | 0.427 | 0.367 | 0.448 | 0.298 | 0.409 | 0.351 | 0.432 | 0.321 | 0.410 |
| | 720 | 0.896 | 0.712 | 0.840 | 0.670 | 0.847 | 0.691 | 0.901 | 0.714 | 1.195 | 0.695 | 0.964 | 0.746 | 0.861 | 0.671 | 0.886 | 0.714 | 0.828 | 0.685 |
| Traffic | 96 | 0.430 | 0.294 | 0.506 | 0.336 | 0.395 | 0.268 | 0.462 | 0.295 | 0.587 | 0.366 | 0.593 | 0.321 | 0.650 | 0.397 | 0.649 | 0.389 | 0.651 | 0.391 |
| | 192 | 0.452 | 0.307 | 0.508 | 0.333 | 0.417 | 0.276 | 0.466 | 0.296 | 0.604 | 0.373 | 0.617 | 0.336 | 0.600 | 0.372 | 0.601 | 0.366 | 0.602 | 0.363 |
| | 336 | 0.470 | 0.316 | 0.518 | 0.335 | 0.433 | 0.283 | 0.482 | 0.304 | 0.621 | 0.383 | 0.629 | 0.336 | 0.606 | 0.374 | 0.609 | 0.369 | 0.609 | 0.366 |
| | 720 | 0.498 | 0.323 | 0.553 | 0.354 | 0.467 | 0.302 | 0.514 | 0.322 | 0.626 | 0.382 | 0.640 | 0.350 | 0.646 | 0.395 | 0.647 | 0.387 | 0.647 | 0.385 |
| Weather | 96 | 0.162 | 0.207 | 0.164 | 0.210 | 0.174 | 0.214 | 0.177 | 0.218 | 0.217 | 0.296 | 0.172 | 0.220 | 0.194 | 0.248 | 0.192 | 0.232 | 0.166 | 0.213 |
| | 192 | 0.210 | 0.250 | 0.214 | 0.252 | 0.221 | 0.254 | 0.225 | 0.259 | 0.276 | 0.336 | 0.219 | 0.261 | 0.234 | 0.290 | 0.240 | 0.271 | 0.213 | 0.254 |
| | 336 | 0.265 | 0.290 | 0.268 | 0.293 | 0.278 | 0.296 | 0.278 | 0.297 | 0.339 | 0.380 | 0.280 | 0.306 | 0.283 | 0.335 | 0.292 | 0.307 | 0.269 | 0.294 |
| | 720 | 0.342 | 0.340 | 0.344 | 0.342 | 0.358 | 0.347 | 0.354 | 0.348 | 0.403 | 0.428 | 0.365 | 0.359 | 0.348 | 0.385 | 0.364 | 0.353 | 0.346 | 0.343 |

## 5.2 Main Results

We present the forecasting results of our FilterNet compared to several representative baselines on eight benchmarks with various prediction lengths in Table 1. Additional results with different lookback window lengths are reported in Appendix F and G.3. Table 1 demonstrates that our model consistently outperforms other baselines across different benchmarks. The average improvement of FilterNet over all baseline models is statistically significant at the confidence of 95%. Specifically, PaiFilter performs well on small datasets (e.g., ETTh1), while TexFilter excels on large datasets (e.g., Electricity) due to the ability to model the more complex and contextual correlations present in larger datasets. Also, the prediction performance of iTransformer [17], which achieves the best results on the Traffic dataset (862 variables) but not on smaller datasets, suggests that simpler structures may be more suitable for smaller datasets, while larger datasets require more contextual structures due to their complex relationships. Compared with FITS [32] built on low-pass filters, our model outperforms it validating an all-pass filter is more effective. Since PaiFilter is simple yet effective, the following FilterNet in the experimental section refer to PaiFilter unless otherwise stated.

## 5.3 Model Analysis

In this part, we conduct experiments to delve into a thorough exploration of frequency filters, including their modeling capabilities, comparisons among different types of frequency filters, and the various factors impacting their performance. Detailed experimental settings are provided in Appendix C.5.

**Modeling Capability of Frequency Filters** Despite the simplicity of frequency filter architecture, Table 1 demonstrates that this architecture can also achieve competitive performance. Hence, in this part, we perform experiments to explore the modeling capability of frequency filters. Particularly, given the significance of trend and seasonal signals in time series forecasting, we investigate the efficacy of simple filters in modeling these aspects. We generate a trend signal and a multi-period signal with noise, and then we leverage the frequency filters (i.e., PaiFilter) to perform training on the two signals respectively. Subsequently, we produce prediction values based on the trained frequency filters. Specifically, Figure 4(a) and 4(b) show that the filter can effectively model trend and periodic signals respectively even compared with state-of-the-art iTransformer [17] when the data contains noise. These results illustrate that the filter has excellent and robust modeling capabilities for trend and periodic signals which are important components for time series. This can also explain the effectiveness of FilterNet since the filters can perform well in such settings.

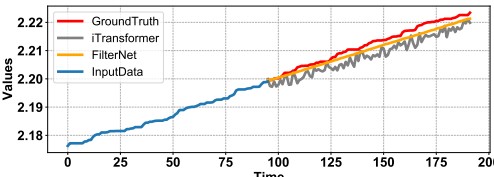 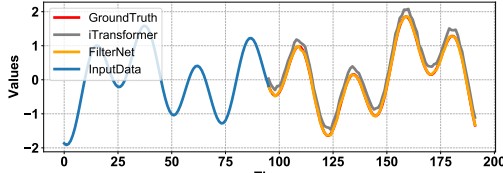

(a) Prediction on trend signals with noises  (b) Prediction on multi-periodic signals with noises

Figure 4: Predictions produced by FilterNet on trend and multi-periodic signals with noises. When adding noises for interference, FilterNet can perform more robust forecasting than iTransformer [17].

Table 2: Performance evaluation of forecasting using two different kinds of frequency filters on the ETTh1 and Exchange datasets with a lookback window size of 96 and the prediction lengths $\tau \in \{96, 192, 336, 720\}$. Results highlighted in red indicate the best performance.

| Datasets | ETTh1 | | | | | | | | Exchange | | | | | | | |
|---|---|---|---|---|---|---|---|---|---|---|---|---|---|---|---|---|
| Lengths | 96 | | 192 | | 336 | | 720 | | 96 | | 192 | | 336 | | 720 | |
| Metrics | MSE | MAE | MSE | MAE | MSE | MAE | MSE | MAE | MSE | MAE | MSE | MAE | MSE | MAE | MSE | MAE |
| $\mathcal{H}_\phi^{(Uni)}$ | 0.375 | 0.394 | 0.436 | 0.422 | 0.476 | 0.443 | 0.474 | 0.469 | 0.083 | 0.202 | 0.174 | 0.296 | 0.326 | 0.413 | 0.840 | 0.670 |
| $\mathcal{H}_\phi^{(Ind)}$ | 0.382 | 0.402 | 0.430 | 0.429 | 0.472 | 0.451 | 0.481 | 0.473 | 0.091 | 0.211 | 0.186 | 0.305 | 0.380 | 0.449 | 0.896 | 0.712 |

**Shared vs. Unique Filters Among Channels** To analyze the different channel strategies of filters, we further conduct experiments on the ETTh and Exchange datasets. Specifically, we compare forecasting performance under different prediction lengths between two different types of frequency filters, i.e., $\mathcal{H}_\phi^{(Uni)}$ and $\mathcal{H}_\phi^{(Ind)}$. In $\mathcal{H}_\phi^{(Uni)}$, filters are shared across different channels, whereas $\mathcal{H}_\phi^{(Ind)}$ signifies filters unique to each channel. The evaluation results are presented in Table 2. It demonstrates that filters shared among different channels consistently outperform across all prediction lengths. In addition, we visualize the prediction values predicted on the ETTh1 dataset by the two different types of filters, as illustrated in Figure 11 (see Appendix G.1). The visualization reveals that the prediction values generated by filters shared among different channels exhibit a better fit than the unique filters. Therefore, the strategy of channel sharing seems to be better suited for time series forecasting and filter designs, which is also validated in DLinear [12] and PatchTST [16].

**Visualization of Prediction** We present a prediction showcase on ETTh1 dataset, as shown in Figure 5. We select iTransformer [17], PatchTST [16] as the representative compared methods. Comparing with these different state-of-the-art models, we can observe FilterNet delivers the most accurate predictions of future series variations, which has demonstrated superior performance. In addition, we include more visualization cases and please refer to Appendix G.3.

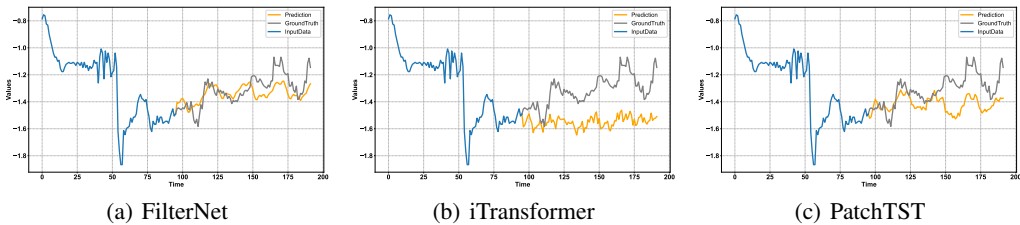

(a) FilterNet  (b) iTransformer  (c) PatchTST

Figure 5: Visualization of prediction on the ETTh1 dataset with lookback and horizon length as 96.

**Visualization of Frequency Filters** To provide a comprehensive overview of the frequency response characteristics of frequency filters, we conduct visualization experiments on the Weather, ETTh, and Traffic datasets with the lookback window length of 96 and the prediction length of 96. The frequency response characteristics of learned filters are visualized in Figure 7. From Figures 7(a) and 7(b), we can observe that compared with Transformer-based approaches (e.g., iTransformer [17], PatchTST [16]) tend to attenuate high-frequency components and preserve low-frequency information, FilterNet exhibits a more nuanced and adaptive filtering behavior that can be capable of attending to all frequency components. Figure 7(c) demonstrates that the main patterns of the Traffic dataset primarily resides in the low-frequency range. This observation also explains why iTransformer performs well on the Traffic dataset, despite its low-frequency nature. Overall, Figure 7 demonstrates

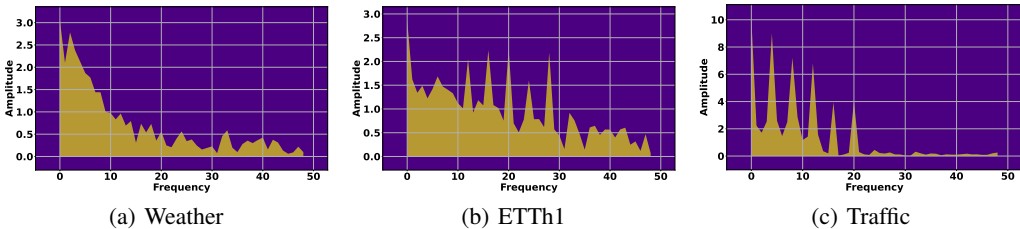

(a) Weather          (b) ETTh1          (c) Traffic

Figure 7: Spectrum visualizations of filters learned on the Weather, ETTh1, and Traffic datasets.

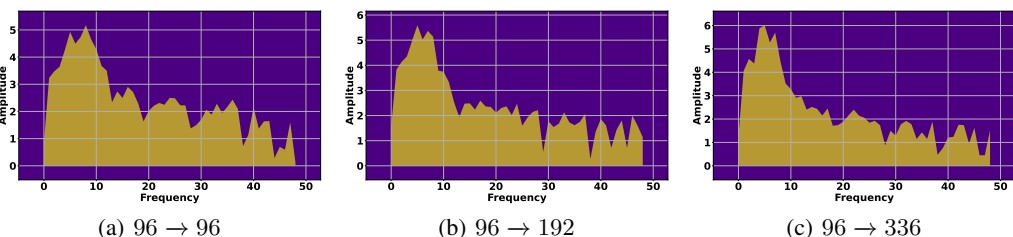

(a) $96 \rightarrow 96$        (b) $96 \rightarrow 192$        (c) $96 \rightarrow 336$

Figure 8: Spectrum visualizations of filters learned on ETTm1 with different prediction lengths.

that FilterNet possesses comprehensive processing capabilities. Moreover, visualization experiments conducted on the ETTm1 dataset across various prediction lengths, as shown in Figure 8, further illustrate the extensive processing abilities of FilterNet. Additional results conducted on different lookback window lengths and prediction lengths can be found in Appendix G.2.

**Efficiency Analysis**    The complexity of FilterNet is $\mathcal{O}(\mathrm{Log}\, L)$ where $L$ is the input length. To comprehensively assess efficiency, we evaluate it based on two dimensions: memory usage and training time. Specifically, we choose two different sizes of datasets: the Exchange (8 variables, 7588 timestamps) and Electricity datasets (321 variables, 26304 timestamps). We compare the efficiency of our FilterNet with the representative Transformer- and MLP-based methods under the same settings (lookback window length of 96 and prediction length of 96), and the results are shown in Figure 6. It highlights that FilterNet surpasses other Transformer models, regardless of dataset size. While our approach exhibits similar efficiency to DLinear, our effective results outperform its performance. In Appendix E, we further conduct ablation studies to validate the rationale of FilterNet designs.

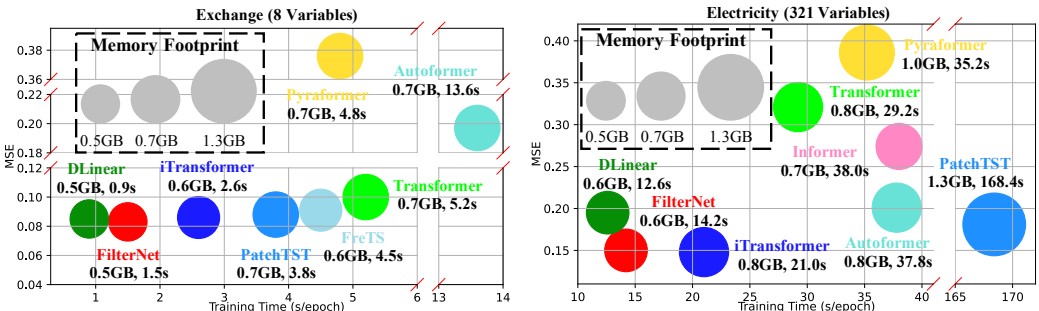

Figure 6: Model effectiveness and efficiency comparison on the Exchange and Electricity datasets.

## 6  Conclusion Remarks

In this paper, we explore an interesting direction from a signal processing perspective and make a new attempt to apply frequency filters directly for time series forecasting. We propose a simple yet effective architecture, *FilterNet*, built upon our proposed two kinds of frequency filters to accomplish the forecasting. Our comprehensive empirical experiments on eight benchmarks have validated the superiority of our proposed method in terms of effectiveness and efficiency. We also include many careful and in-depth model analyses of FilterNet and the internal filters, which demonstrate many good properties. We hope this work can facilitate more future research integrating signal processing techniques or filtering processes with deep learning on time series modeling and accurate forecasting.

## Acknowledgments and Disclosure of Funding

The work was supported in part by the National Natural Science Foundation of China under Grant 92367110, the Shanghai Baiyulan TalentPlan Pujiang Project under Grant 23PJ1413800, and Shanxi Scholarship Council of China (2024-61).

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

# A   More Analysis about the Architecture of FilterNet

**The Necessity of Instance Normalization Block**   From the frequency perspective, the mean value is equal to zero frequency component. Specifically, given a signal $x[n]$ with a length of $N$, we can obtain its corresponding discrete Fourier transform $\mathcal{X}[k]$ by:

$$\mathcal{X}[k] = \frac{1}{N} \sum_{n=0}^{N-1} x[n] e^{2\pi jnk/N} \tag{10}$$

where $j$ is the imaginary unit. We set $k$ as 0 and then,

$$\mathcal{X}[0] = \frac{1}{N} \sum_{n=0}^{N-1} x[n] e^{2\pi jn0/N}$$
$$= \frac{1}{N} \sum_{n=0}^{N-1} x[n]. \tag{11}$$

According the above equation, we can find that the mean value $\frac{1}{N} \sum_{n=0}^{N-1} x[n]$ in the time domain is equal to the zero frequency component $\mathcal{X}[0]$ in the frequency domain. Similarly, we can also view the standard deviation from the frequency perspective, and it is related to the power spectral density.

According to these analysis, the instance normalization is analogous to a form of data preprocessing. Given that filters are primarily crafted to discern particular patterns within input data, while instance normalization aims to normalize each instance in a dataset, a function distinct from the conventional role of filters, we treat instance normalization as a distinct block within our FilterNet architecture.

**Frequency Filter Block**   Recently, Transformer- and MLP-based methods have emerged as the two main paradigms for time series forecasting, exhibiting competitive performance compared to other model architectures. Building on prior work that conceptualizes self-attention and MLP architectures as forms of global convolution [35, 13], it becomes apparent that frequency filters hold promise for time series forecasting tasks. Just as self-attention mechanisms capture global dependencies and MLPs learn to convolve features across the entire input space, frequency filters offer a means to extract and emphasize specific temporal patterns and trends from time series data. By applying frequency filters to time series data, we can learn recurring patterns, trends, and periodic behaviors that are essential for forecasting future time series data and making accurate predictions.

**Feed-forward Network**   Incorporating a feed-forward network within the FilterNet architecture is essential for enhancing the model's capacity to capture complex relationships and non-linear patterns within the data. While frequency filters excel at extracting specific frequency components and temporal patterns from time series data, they may not fully capture the intricate dependencies and higher-order interactions present in real-world datasets [17]. By integrating a feed-forward network, the model gains the ability to learn hierarchical representations and abstract patterns from the input data, allowing it to capture more nuanced relationships and make more accurate predictions. This combination of frequency filters and a feed-forward network leverages the strengths of both approaches, enabling the model to effectively process and analyze time series data across different frequency bands while accommodating the diverse and often nonlinear nature of temporal dynamics. Overall, the inclusion of a feed-forward network enriches the expressive power of FilterNet, leading to improved forecasting performance and robustness across various domains.

# B   Explanations about the Design of Two Filters

Self-attention mechanism is a highly data-dependent operation that both derives its parameters from data and subsequently applies these parameters back to the data. Concretely, given the input data $X$, the self-attention can be formulated as:

$$\mathrm{SA}(Q, K, V) = \mathrm{softmax}\left(\frac{QK^T}{\sqrt{d_k}}\right)V, \tag{12}$$

where $Q$ (queries), $K$ (keys), and $V$ (values) are linear transformations of the input data X, as:

$$Q = W_Q X, K = W_K X, V = W_V X, \tag{13}$$

where $W_Q$, $W_K$, and $W_V$ are learned weight matrices. Since the values $Q$, $K$, and $V$ are derived directly from the input data $X$, the attention scores SA are inherently dependent on the data.

Unlike self-attention mechanism that dynamically adapts to the input data during inference, MLPs maintain a consistent architecture regardless of the dataset characteristics. Specifically, for the input data $X$, MLP can be formulated as:

$$\text{MLP}(X) = WX + b, \tag{14}$$

where $W$ are the learned weights and $b$ are the learned biases. Once trained, the weights $W$ and biases $b$ remain static, meaning that they do not dynamically change in response to new data inputs.

MLPs are straightforward, less data-dependent models that apply fixed transformations to the input data, making them suitable for tasks with static relationships between the input data. In contrast, self-attention mechanisms are highly data-dependent, dynamically computing attention scores based on the input data to capture complex, context-specific dependencies, making them ideal for tasks requiring an understanding of sequential or structured data.

Inspired by the two paradigms, FilterNet designs two corresponding types of filters: plain shaping filters and contextual shaping filters. Plain shaping filters offer stability and efficiency, making them suitable for tasks with static relationships. In contrast, contextual shaping filters provide the flexibility to capture dynamic dependencies, excelling in tasks that require context-sensitive analysis. This dual approach allows FilterNet to effectively handle a wide range of data types and forecasting scenarios, combining the best aspects of both paradigms to achieve superior performance.

## C  Experimental Details

### C.1  Datasets

We evaluate the performance of our proposed FilterNet on eight popular datasets, including Exchange, Weather, Traffic, Electricity, and ETT datasets. In detail, the Exchange[2] dataset collects daily exchange rates of eight different countries including Singapore, Australia, British, Canada, Switzerland, China, Japan, and New Zealand ranging from 1990 to 2016. The Weather[3] dataset, including 21 meteorological indicators such as air temperature and humidity, is collected every 10 minutes from the Weather Station of the Max Planck Institute for Biogeochemistry in 2020. The Traffic [15] dataset contains hourly traffic data measured by 862 sensors on San Francisco Bay area freeways, which has been collected since January 1, 2015. The Electricity[4] dataset collects the hourly electricity consumption of 321 clients from 2012 to 2014. The ETT[5] (Electricity Transformer Temperature) datasets contain two visions of the sub-dataset: ETTh and ETTm, collected from electricity transformers every 15 minutes and 1 hour between July 2016 and July 2018. Thus, in total we have 4 ETT datasets (ETTm1, ETTm2, ETTh1, and ETTh2) recording 7 features such as load and oil temperature. The details about these datasets are summarized in Table 3.

Table 3: The details of datasets.

| Datasets | ETTh1 | ETTh2 | ETTm1 | ETTm2 | Electricity | Weather | Traffic | Exchange |
|---|---|---|---|---|---|---|---|---|
| Variables | 7 | 7 | 7 | 7 | 321 | 21 | 862 | 8 |
| Timesteps | 17420 | 17420 | 69680 | 69680 | 26304 | 52696 | 17544 | 7588 |
| Frequency | Hourly | Hourly | 15min | 15min | Hourly | 10min | Hourly | Daily |
| Information | Electricity | Electricity | Electricity | Electricity | Electricity | Weather | Traffic | Economy |

### C.2  Baselines

We choose twelve well-acknowledged and state-of-the-art models for comparison to evaluate the effectiveness of our proposed FilterNet for time series forecasting, including Frequency-based models, TCN-based models, MLP-based models, and Transformer-based models. We introduce these models as follows:

---

[2]https://github.com/laiguokun/multivariate-time-series-data

[3]https://www.bgc-jena.mpg.de/wetter/

[4]https://archive.ics.uci.edu/ml/datasets/ElectricityLoadDiagrams20112014

[5]https://github.com/zhouhaoyi/ETDataset

**FreTS** [13] introduces a novel direction of applying MLPs in the frequency domain to effectively capture the underlying patterns of time series, benefiting from global view and energy compaction. The official implementation is available at `https://github.com/aikunyi/FreTS`. To ensure fair and objective comparison, the results showed in Table 4 are obtained using instance normalization instead of min-max normalization in the original code.

**FITS** [32] performs time series analysis through interpolation in the complex frequency domain, enjoying low cost with 10k parameters. The official implementation is available at `https://github.com/VEWOXIC/FITS`. Because its original paper doesn't provide the forecasting results with the fixed lookback length $L = 96$, we report the performance of FITS with lookback length $L = 96$ under five runs in Table 1.

**MICN** [39] employs multi-scale branch structure to model different potential patterns separately with linear complexity. It combines local features and global correlations to capture the overall view of time series. The official implementation is available at `https://github.com/wanghq21/MICN`. The experimental results showed in Table 4 follow its original paper.

**TimesNet** [40] transforms 1D time series into a set of 2D tensors based on multiple periods to analyse temporal variations. The above transformation allows the 2D-variations to be easily captured by 2D kernels with encoding the intraperiod- and interperiod-variations into the columns and rows of the 2D tensors respectively. The official implementation is available at `https://github.com/thuml/TimesNet`. The results showed in Table 1 follow iTransformer [17] and the results showed in Table 5 follow RLinear [23].

**DLinear** [12] utilizes a simple yet effective one-layer linear model to capture temporal relationships between input and output sequences. The official implementation is available at `https://github.com/cure-lab/LTSF-Linear`. We report the performance of DLinear with lookback length $L \in \{96, 336\}$ under five runs in Table 1 and 5.

**RLinear** [23] uses linear mapping to model periodic features in multivariate time series with robustness for diverse periods when increasing the input length. It applies RevIN (reversible normalization) and CI (Channel Independent) to improve overall forecasting performance by simplifying learning about periodicity. The official implementation is available at `https://github.com/plumprc/RTSF`. The experimental results showed in Table 1 follow iTransformer [17].

**Informer** [14] enhances Transformer with KL-divergence based ProbSparse attention for $O(L \log L)$ complexity, efficiently encoding dependencies among variables and introducing a novel architecture with a DMS forecasting strategy. The official implementation is available at `https://github.com/zhouhaoyi/Informer2020` and the experimental results showed in Table 4 follow FEDformer [25].

**Autoformer** [15] employs a deep decomposition architecture with auto-correlation mechanism to extract seasonal and trend components from input series, embedding the series decomposition block as an inner operator. The official implementation is available at `https://github.com/thuml/Autoformer` and the experimental results showed in Table 4 follow FEDformer [25].

**Pyraformer** [26] introduces pyramidal attention module (PAM) with an $O(L)$ time and memory complexity where the inter-scale tree structure summarizes features at different resolutions and the intra-scale neighboring connections model the temporal dependencies of different ranges. The official implementation is available at `https://github.com/ant-research/Pyraformer` and the experimental results showed in Table 4 follow DLinear [12].

**FEDformer** [25] implements sparse attention with low-rank approximation in frequency domain, enjoying linear computational complexity and memory cost. And it proposes mixture of experts decomposition to control the distribution shifting. The official implementation is available at `https://github.com/MAZiqing/FEDformer` and the experimental results showed in Table 1 follow iTransformer [17].

**PatchTST** [16] divides time series data into subseries-level patches to extract local semantic information and adopts channel-independence strategy where each channel shares the same embedding and Transformer weights across all the series. The official implementation is available at `https://github.com/yuqinie98/PatchTST`. The experimental results showed in Table 1 follow iTransformer [17]. And because iTransformer doesn't provide the prediction results with lookback length $L = 336$, we report the performance of PatchTST with lookback length $L = 336$ under five runs in Table 5.

**iTransformer** [17] inverts the structure of Transformer without modifying any existing modules by encoding individual series into variate tokens. These tokens are utilized by the attention mechanism to capture multivariate correlations and FFNs are adopted for each variate token to learn nonlinear representations. The official implementation is available at `https://github.com/thuml/iTransformer`. The experimental results showed in Table 1 follow its original paper. And because iTransformer doesn't provide the prediction results with lookback length $L = 336$, we report the performance of iTransformer with lookback length $L = 336$ under five runs in Table 5.

### C.3 Implementation Details

The architecture of our FilterNet is very simple and has two main hyperparameters, i.e., the bandwidth of filters and the hidden size of FFN. As shown in Figure 9, the bandwidth of the filters corresponds to the lookback window length, so we select the lookback window length as the bandwidth accordingly. For the hidden size of FFN, we carefully tune the size over {64, 128, 256, 512}. Following the previous methods [16, 32], we use RevIN [19] as our instance normalization block. Besides, we carefully tune the hyperparameters including the batch size and learning rate on the validation set, and we choose the settings with the best performance. We tune the batch size over {4, 8, 16, 32} and tune the learning rate over {0.01, 0.05, 0.001, 0.005, 0.0001, 0.0005}.

### C.4 Experimental Settings for Simulation Experiments

To evaluate the Transformer's modeling ability for different frequency spectrums, we generate a signal consisting of three different frequency components, i.e., low-frequency, middle-frequency, and high-frequency. We then apply both iTransformer and FilterNet to this signal, respectively, and compare the forecasting results with the ground truth. The results are presented in Figure 1.

### C.5 Experimental Settings for Filters Analysis

**Modeling capability of frequency filters**   We generate two signals: a trend signal with Gaussian noise and a multi-periodic signal with Gaussian noise. We then apply $\mathrm{PaiFilter}$ to these signals with a lookback window length of 96 and a prediction length of 96. The results are displayed in Figure 4.

**Visualization of Frequency Filters**   Given a filter $\mathbf{H} \in \mathbb{R}^{1 \times L}$, where $L$ is its bandwidth, we visualize the frequency response characteristic of the filter by plotting the values in $\mathbb{R}^{1 \times L}$. First, we perform a Fourier transform on these values to obtain the spectrum, which includes the frequency and its corresponding amplitude. Finally, we visualize the spectrum, as shown in Figures 7, 8, and 12.

## D   Study of the Bandwidth of Frequency Filters

The bandwidth parameter (i.e., $L$ in Equation (8) and $D$ in Equation (9)) holds significant importance in the functionality of filters. In this part, we conduct experiments on the Weather dataset to delve into the impact of bandwidth on forecasting performance. We explore a range of bandwidth values within the set $\{96, 128, 192, 256, 320, 386, 448, 512\}$ while keeping the lookback window length and prediction length constant. Specifically, we conduct experiments to evaluate the impact under three different combinations of lookback window length and prediction length, i.e., $96 \rightarrow 96$, $96 \rightarrow 192$, and $192 \rightarrow 192$, and the results are represented in Figure 9. We observe clear trends in the relationship between bandwidth settings and lookback window length. Figure 9(a) and Figure 9(b) show that increasing the bandwidth results in minimal changes in forecasting performance. Figure 9(c) demonstrates that while forecasting performance fluctuates with increasing bandwidth, it is optimal when the bandwidth equals the lookback window length. These results indicate that using the lookback window length as the bandwidth is sufficient since the filters can effectively model the data at this setting, and it also results in lower complexity.

## E   Ablation Study

To validate the rationale behind the architectural design of our FilterNet, we conduct ablation studies on the ETTm1, ETTh1, and Electricity datasets. We evaluate the impact on the model's performance

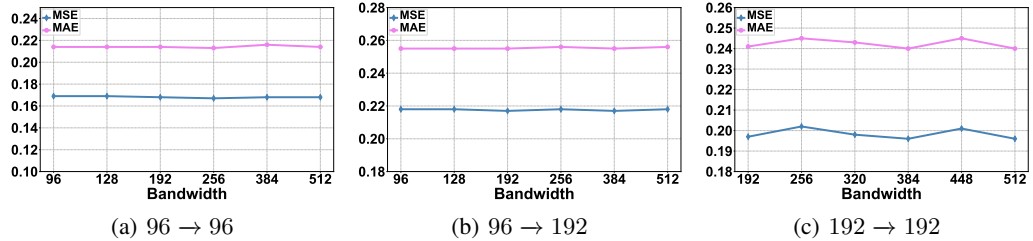

(a) $96 \rightarrow 96$        (b) $96 \rightarrow 192$        (c) $192 \rightarrow 192$

Figure 9: MSE and MAE of filters under different bandwidths on the Weather dataset.

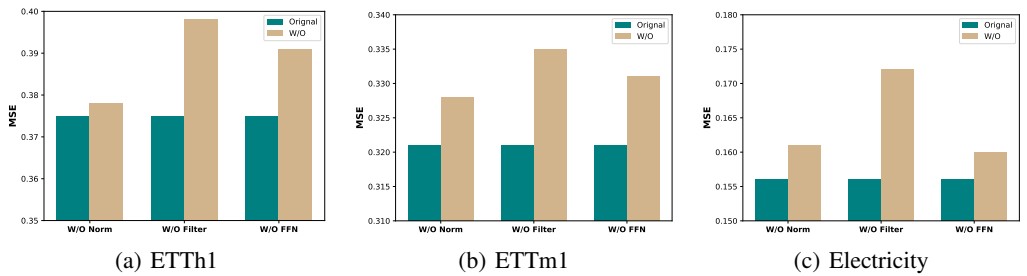

(a) ETTh1        (b) ETTm1        (c) Electricity

Figure 10: Ablation Studies on the ETTh1, ETTm1, and Electricity datasets.

by eliminating the particular component of the FilterNet architecture. The evaluation results are present in Figure 10. In the figure, W/O Norm indicates that instance normalization and inverse instance normalization have been removed from FilterNet. W/O Filter signifies the removal of the filter block, and W/O FFN denotes the exclusion of the feed-forward network. The experiments are conducted with lookback window length of 96 and output length of 96. From the figure, we can conclude that each block is indispensable, as the removal of any component results in a noticeable decrease in performance. This highlights the critical role each block plays in the overall architecture of FilterNet, contributing to its effectiveness in time series forecasting.

# F   Additional Results

Table 4 compares the performance of various methods with our FilterNet, demonstrating that our model consistently outperforms the others. To further assess the performance of FilterNet under different lookback window lengths, we conducted experiments on the ETTh1, ETTm1, Exchange, Weather, and Electricity datasets with the lookback window length of 336. The results, shown in Table 5, indicate that our model achieves the best performance across these datasets.

Table 4: Multivariate long-term forecasting results with prediction lengths $\tau \in \{96, 192, 336, 720\}$ and fixed lookback length $L = 96$. The best results are in red and the second best are in blue.

| Models | | **TexFilter** | | **PaiFilter** | | Autoformer | | Informer | | Pyraformer | | MICN | | FreTS | |
|---|---|---|---|---|---|---|---|---|---|---|---|---|---|---|---|
| Metrics | | MSE | MAE | MSE | MAE | MSE | MAE | MSE | MAE | MSE | MAE | MSE | MAE | MSE | MAE |
| ETTm1 | 96 | 0.321 | **0.361** | **0.318** | **0.358** | 0.505 | 0.475 | 0.672 | 0.571 | 0.543 | 0.510 | **0.316** | 0.362 | 0.335 | 0.372 |
| | 192 | 0.367 | **0.387** | **0.364** | **0.383** | 0.553 | 0.496 | 0.795 | 0.669 | 0.557 | 0.537 | **0.363** | 0.390 | 0.388 | 0.401 |
| | 336 | **0.401** | **0.409** | **0.396** | **0.406** | 0.621 | 0.537 | 1.212 | 0.871 | 0.754 | 0.655 | 0.408 | 0.426 | 0.421 | 0.426 |
| | 720 | **0.477** | **0.448** | **0.456** | **0.444** | 0.671 | 0.561 | 1.166 | 0.823 | 0.908 | 0.724 | 0.481 | 0.476 | 0.486 | 0.465 |
| ETTm2 | 96 | **0.175** | **0.258** | **0.174** | **0.257** | 0.255 | 0.339 | 0.365 | 0.453 | 0.435 | 0.507 | 0.179 | 0.275 | 0.189 | 0.277 |
| | 192 | **0.240** | **0.301** | **0.240** | **0.300** | 0.281 | 0.340 | 0.533 | 0.563 | 0.730 | 0.673 | 0.307 | 0.376 | **0.258** | 0.326 |
| | 336 | **0.311** | **0.347** | **0.297** | **0.339** | 0.339 | 0.372 | 1.363 | 0.887 | 1.201 | 0.845 | 0.325 | 0.388 | 0.343 | 0.390 |
| | 720 | **0.414** | **0.405** | **0.392** | **0.393** | 0.422 | 0.419 | 3.379 | 1.338 | 3.625 | 1.451 | 0.502 | 0.490 | 0.495 | 0.480 |
| ETTh1 | 96 | **0.382** | **0.402** | **0.375** | **0.394** | 0.449 | 0.459 | 0.865 | 0.713 | 0.664 | 0.612 | 0.421 | 0.431 | 0.395 | 0.407 |
| | 192 | **0.430** | **0.429** | **0.436** | **0.422** | 0.500 | 0.482 | 1.008 | 0.792 | 0.790 | 0.681 | 0.474 | 0.487 | 0.448 | 0.440 |
| | 336 | **0.472** | **0.451** | **0.476** | **0.443** | 0.521 | 0.496 | 1.107 | 0.809 | 0.891 | 0.738 | 0.569 | 0.551 | 0.499 | 0.472 |
| | 720 | **0.481** | **0.473** | **0.474** | **0.469** | 0.514 | 0.512 | 1.181 | 0.865 | 0.963 | 0.782 | 0.770 | 0.672 | 0.558 | 0.532 |
| ETTh2 | 96 | **0.293** | **0.343** | **0.292** | **0.343** | 0.358 | 0.397 | 3.755 | 1.525 | 0.645 | 0.597 | 0.299 | **0.364** | 0.309 | **0.364** |
| | 192 | **0.374** | **0.396** | **0.369** | **0.395** | 0.456 | 0.452 | 5.602 | 1.931 | 0.788 | 0.683 | 0.441 | 0.454 | 0.395 | 0.425 |
| | 336 | **0.417** | **0.430** | **0.420** | **0.432** | 0.482 | 0.486 | 4.721 | 1.835 | 0.907 | 0.747 | 0.654 | 0.567 | 0.462 | 0.467 |
| | 720 | **0.449** | **0.460** | **0.430** | **0.446** | 0.515 | 0.511 | 3.647 | 1.625 | 0.963 | 0.783 | 0.956 | 0.716 | 0.721 | 0.604 |
| ECL | 96 | **0.147** | **0.245** | 0.176 | 0.264 | 0.201 | 0.317 | 0.274 | 0.368 | 0.386 | 0.449 | **0.164** | 0.269 | 0.176 | **0.258** |
| | 192 | **0.160** | **0.250** | 0.185 | 0.270 | 0.222 | 0.334 | 0.296 | 0.386 | 0.386 | 0.443 | 0.177 | 0.285 | **0.175** | **0.262** |
| | 336 | **0.173** | **0.267** | 0.202 | 0.286 | 0.231 | 0.338 | 0.300 | 0.394 | 0.378 | 0.443 | 0.193 | 0.304 | **0.185** | **0.278** |
| | 720 | **0.210** | **0.309** | 0.242 | 0.319 | 0.254 | 0.361 | 0.373 | 0.439 | 0.376 | 0.445 | **0.212** | 0.321 | 0.220 | **0.315** |
| Exchange | 96 | **0.091** | **0.211** | **0.083** | **0.202** | 0.197 | 0.323 | 0.847 | 0.752 | 0.376 | 1.105 | 0.102 | 0.235 | **0.091** | 0.217 |
| | 192 | 0.186 | **0.305** | **0.174** | **0.296** | 0.300 | 0.369 | 1.204 | 0.895 | 1.748 | 1.151 | **0.172** | 0.316 | 0.175 | 0.310 |
| | 336 | 0.380 | 0.449 | **0.326** | **0.413** | 0.509 | 0.524 | 1.672 | 1.036 | 1.874 | 1.172 | **0.272** | **0.407** | 0.334 | 0.434 |
| | 720 | 0.896 | 0.712 | 0.840 | **0.670** | 1.447 | 0.941 | 2.478 | 1.310 | 1.943 | 1.206 | **0.714** | **0.658** | **0.716** | 0.674 |
| Traffic | 96 | **0.430** | **0.294** | **0.506** | 0.336 | 0.613 | 0.388 | 0.719 | 0.391 | 2.085 | 0.468 | 0.519 | **0.309** | 0.593 | 0.378 |
| | 192 | **0.452** | **0.307** | **0.508** | 0.333 | 0.616 | 0.382 | 0.696 | 0.379 | 0.867 | 0.467 | 0.537 | **0.315** | 0.595 | 0.377 |
| | 336 | **0.470** | **0.316** | **0.518** | 0.335 | 0.622 | 0.337 | 0.777 | 0.420 | 0.869 | 0.469 | 0.534 | **0.313** | 0.609 | 0.385 |
| | 720 | **0.498** | **0.323** | **0.553** | 0.354 | 0.660 | 0.408 | 0.864 | 0.472 | 0.881 | 0.473 | 0.577 | **0.325** | 0.673 | 0.418 |
| Weather | 96 | **0.162** | **0.207** | 0.164 | 0.210 | 0.266 | 0.336 | 0.300 | 0.384 | 0.896 | 0.556 | **0.161** | 0.229 | 0.174 | **0.208** |
| | 192 | **0.210** | **0.250** | **0.214** | **0.252** | 0.307 | 0.367 | 0.598 | 0.544 | 0.622 | 0.624 | 0.220 | 0.281 | 0.219 | **0.250** |
| | 336 | **0.265** | **0.290** | **0.268** | **0.293** | 0.359 | 0.395 | 0.578 | 0.523 | 0.739 | 0.753 | 0.278 | 0.331 | 0.273 | **0.290** |
| | 720 | 0.342 | **0.340** | 0.344 | 0.342 | 0.419 | 0.428 | 1.059 | 0.741 | 1.004 | 0.934 | **0.311** | 0.356 | **0.334** | **0.332** |

Table 5: Time series forecasting comparison. We set the lookback window size $L$ as 336 and the prediction length as $\tau \in \{96, 192, 336, 720\}$. The best results are in red and the second best are in blue.

| Models | | **PaiFilter** | | DLinear | | iTransformer | | PatchTST | | TimesNet | |
|---|---|---|---|---|---|---|---|---|---|---|---|
| Metrics | | MSE | MAE | MSE | MAE | MSE | MAE | MSE | MAE | MSE | MAE |
| ETTh1 | 96 | **0.379** | **0.404** | 0.384 | **0.405** | 0.402 | 0.418 | **0.381** | **0.405** | 0.398 | 0.418 |
| | 192 | **0.417** | **0.428** | **0.430** | **0.442** | 0.450 | 0.449 | 0.442 | 0.446 | 0.447 | 0.449 |
| | 336 | **0.437** | **0.443** | 0.447 | **0.448** | 0.479 | 0.470 | **0.445** | 0.454 | 0.493 | 0.468 |
| | 720 | **0.458** | **0.472** | 0.504 | 0.515 | 0.584 | 0.548 | **0.490** | **0.493** | 0.518 | 0.504 |
| ETTm1 | 96 | **0.289** | **0.344** | 0.300 | **0.345** | 0.303 | 0.357 | **0.294** | **0.345** | 0.335 | 0.380 |
| | 192 | **0.331** | **0.369** | 0.336 | **0.366** | 0.345 | 0.383 | **0.334** | 0.371 | 0.358 | 0.388 |
| | 336 | **0.364** | **0.389** | 0.372 | **0.390** | 0.382 | 0.405 | **0.371** | 0.392 | 0.406 | 0.418 |
| | 720 | **0.425** | **0.423** | 0.427 | **0.423** | 0.443 | 0.439 | **0.421** | **0.419** | 0.449 | 0.443 |
| Exchange | 96 | **0.087** | 0.216 | **0.085** | **0.209** | 0.099 | 0.226 | 0.093 | **0.213** | 0.117 | 0.253 |
| | 192 | **0.163** | **0.301** | **0.162** | **0.296** | 0.216 | 0.337 | 0.194 | 0.315 | 0.298 | 0.410 |
| | 336 | **0.287** | **0.399** | **0.350** | 0.445 | 0.395 | 0.466 | 0.354 | **0.435** | 0.456 | 0.513 |
| | 720 | **0.413** | **0.492** | **0.898** | 0.725 | 0.962 | 0.745 | 0.903 | **0.712** | 1.608 | 0.961 |
| Weather | 96 | **0.150** | **0.183** | 0.175 | 0.235 | 0.164 | 0.216 | **0.151** | **0.197** | 0.172 | 0.220 |
| | 192 | **0.193** | **0.221** | 0.218 | 0.278 | 0.205 | 0.251 | **0.197** | **0.244** | 0.219 | 0.261 |
| | 336 | **0.246** | **0.258** | 0.263 | 0.314 | 0.256 | 0.290 | **0.251** | **0.285** | 0.280 | 0.306 |
| | 720 | **0.308** | **0.295** | 0.324 | 0.362 | 0.326 | 0.338 | **0.321** | **0.335** | 0.365 | 0.359 |
| Electricity | 96 | **0.132** | **0.224** | 0.140 | 0.237 | 0.133 | 0.229 | **0.130** | **0.222** | 0.168 | 0.272 |
| | 192 | **0.143** | **0.237** | 0.154 | 0.250 | 0.156 | 0.251 | **0.148** | **0.240** | 0.184 | 0.289 |
| | 336 | **0.155** | **0.253** | 0.169 | 0.268 | 0.172 | 0.267 | **0.167** | **0.261** | 0.198 | 0.300 |
| | 720 | **0.195** | **0.292** | 0.204 | 0.300 | 0.209 | 0.304 | **0.202** | **0.291** | 0.220 | 0.320 |

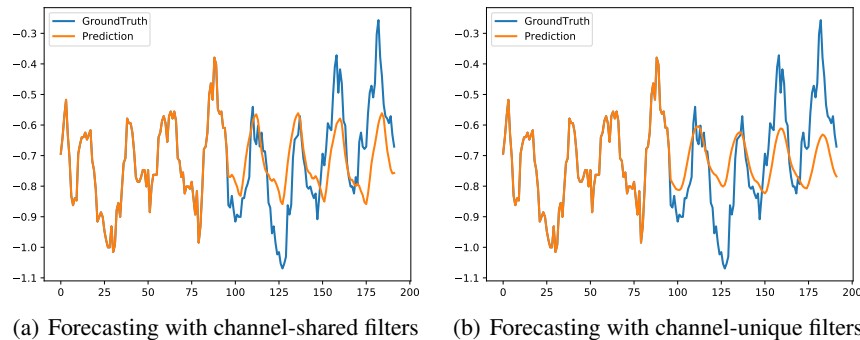

(a) Forecasting with channel-shared filters     (b) Forecasting with channel-unique filters

Figure 11: Visualizations on the ETTh1 dataset.

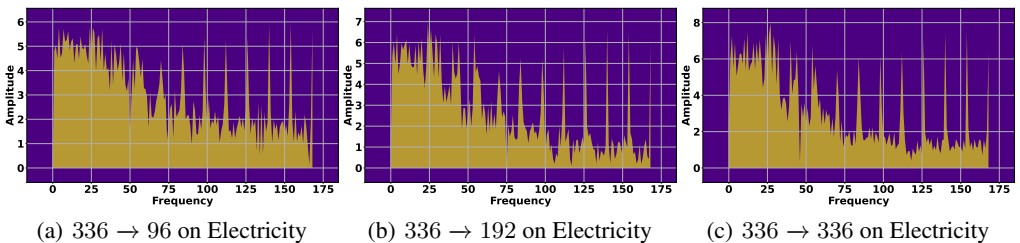

(a) $336 \rightarrow 96$ on Electricity     (b) $336 \rightarrow 192$ on Electricity     (c) $336 \rightarrow 336$ on Electricity

Figure 12: Spectrum visualizations of the filters learned on the Electricity dataset with lookback window length of 336 and prediction lengths $\tau \in \{96, 192, 336\}$.

## G    Visualizations

### G.1    Visualization of Channel-shared vs Channel-unique Filters

To further compare the channel-shared and channel-unique filters, we visualize the prediction values by the corresponding filters. The results are shown in Figure 11. The figure demonstrates that the values predicted by channel-shared filters closely align with the ground truth compared to those predicted by channel-unique filters. This observation is consistent with the findings presented in Table 2, indicating the superiority of channel-shared filters.

### G.2    Visualization of Frequency Filters

We further conduct visualization experiments to explore the learnable filters under different lookback window lengths and prediction lengths. The experiments are performed on the Electricity dataset, and the results are illustrated in Figure 12. These figures illustrate that FilterNet possesses full spectrum learning capability, as the learnable filters exhibit values across the entire spectrum. Besides, we observe that the frequency primarily concentrates in the low and middle ranges which explains that some works based on low-pass filters can also achieve good performance.

### G.3    Visualizations of Predictions

To further offer an evident comparison of our model with the state-of-the-art models, we present supplementary prediction showcases on ETTm1 dataset, and the results are shown in 13. We choose the following representative models, including iTransformer [17], PatchTST [16], and DLinear [12], as the baselines. Comparing with these different types of state-of-the-art models, FilterNet delivers the most accurate predictions of future series variations, demonstrating superior performance.

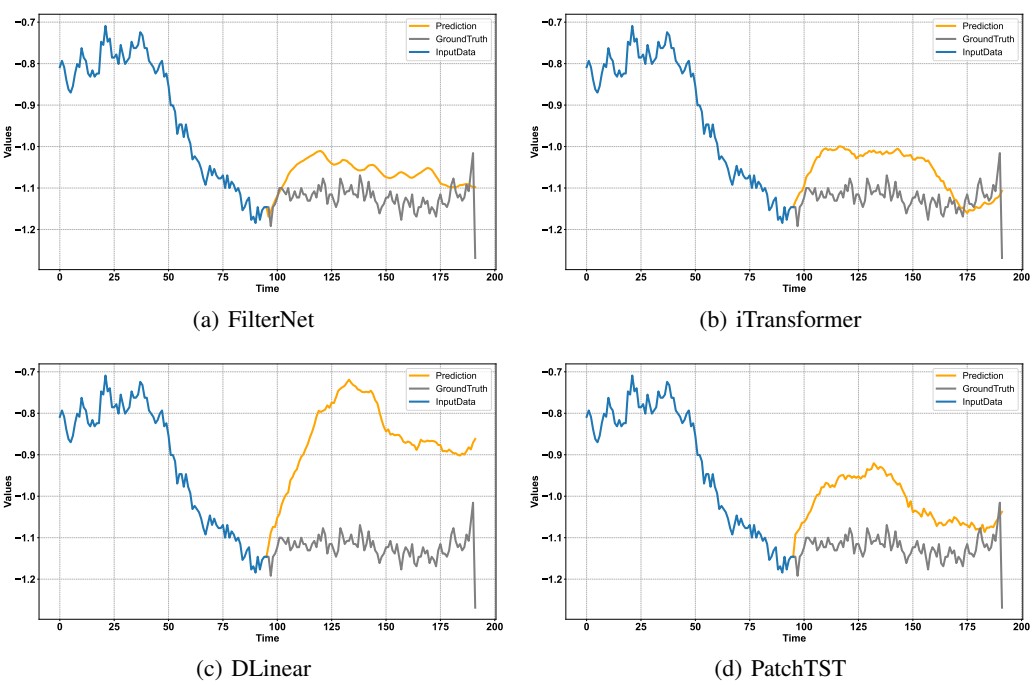

(a) FilterNet

(b) iTransformer

(c) DLinear

(d) PatchTST

Figure 13: Visualization of forecasting results on the ETTm1 dataset with lookback window length $L = 96$ and prediction length $\tau = 96$.

