# OpenReview forum: "FilterNet: Harnessing Frequency Filters for Time Series Forecasting"
_NeurIPS.cc/2024/Conference — NeurIPS 2024 poster_

### Official Review · Reviewer_t3Wx · 2024-07-08

**Soundness:** 4
**Presentation:** 3
**Contribution:** 4
**Rating:** 7
**Confidence:** 4

**Summary:**

This paper introduces FilterNet, a novel neural network architecture for time series forecasting that leverages learnable frequency filters inspired by signal processing techniques. This paper proposes two types of filters within FilterNet: a plain shaping filter that uses a universal frequency kernel for signal filtering and temporal modeling, and a contextual shaping filter that examines filtered frequencies for compatibility with input signals to learn dependencies. This paper demonstrates that FilterNet can effectively handle high-frequency noise and utilize the full frequency spectrum, addressing key limitations of existing Transformer-based forecasters. This paper shows through extensive experiments on eight benchmarks that FilterNet achieves superior performance in both effectiveness and efficiency compared to state-of-the-art methods in time series forecasting.

**Strengths:**

1. This paper studies an interesting research direction, which is to adapt the advanced techniques in the signal processing domain into the time series forecasting, rather than following the existing transformer-based time series works. This is the original and novel contribution to the time series community.
2. The main model part is overall descent. The simple but effective architectures built upon the frequency filters are proposed. I notice there is not many redundant components in the basic model, compared with existing transformer models. It can somehow demonstrates the effectiveness of their proposed two filters.
3. The experiments are solid and convincing. Several experimental results are reported on existing benchmark datasets. Also, some experiments on synthetic periodic and trend signals with noises shown in experiments make sense. The efficiency analysis and visualization is intuitive.

**Weaknesses:**

1. The paper mainly introduces two filters. The connection and difference between the two filters is not highlighted, which makes me curious about which one is better.
2. As far as I know, there exist many frequency filters in signal processing. The motivation for these two filters is not very clear.
3. In experiment part, the results of TexFilter on ETTm1, ETTm2, ETTh1 and Exchange are neither the best nor the second best. Does it show the filter cannot win in many cases of forecasting? Are there any analysis or explanation about it?

**Questions:**

1. Can the authors elaborate on the connection and difference of the two filters?
2. Can the authors give more details about proposed filters?
3. Can the authors explain the lower performance of TexFilter in some cases?

**Limitations:**

See weakness.

---

> ### Author Rebuttal · Authors · 2024-08-07
>
> We appreciate your review and the positive comments regarding our paper. We would like to respond to your comments as follows, and we hope that we address all your concerns below:
>
> **W1. The paper mainly introduces two filters. The connection and difference between the two filters is not highlighted, which makes me curious about which one is better.**
>
> A1. In the literature, compared with other model architectures, MLP-based and Transformer-based methods have achieved competitive performance.
> Inspired by the two paradigms, we design two corresponding types of filters. i.e., PaiFilter and TexFilter.
> PaiFilter offer stability and efficiency, making them
> suitable for tasks with static relationships. In contrast, TexFilter provides the flexibility
> to capture dynamic dependencies, excelling in tasks that require context-sensitive analysis.
> **As shown in Table 1**, PaiFilter performs well on small datasets (e.g., ETTh1), while TexFilter excels on large datasets
> (e.g., Traffic) due to the ability to model the more complex and contextual correlations present in
> larger datasets.
> This dual approach allows FilterNet to effectively handle a wide range of data types and forecasting scenarios,
> combining the best aspects of both paradigms to achieve superior performance.
>
> **In Appendix, we have analyzed the two filters, and introduced the connections (Frequency Filter Block in Appendix A) and differences between them (Explanations about the Design of Two Filters in Appendix B)**.
>
>
> **W2. As far as I know, there exist many frequency filters in signal processing. The motivation for these two filters is not very clear.**
>
> A2. Please refer to A2 in Response to Reviewer ufqs.
>
> **W3. In experiment part, the results of TexFilter on ETTm1, ETTm2, ETTh1 and Exchange are neither the best nor the second best. Does it show the filter cannot win in many cases of forecasting? Are there any analysis or explanation about it?**
>
> A3. Inspired by self-attention mechanisms that are highly data-dependent, dynamically computing attention scores based
> on the input data, TexFilter is designed to learn the filter parameters generated from the input data.
> This makes TexFilter particularly suitable for the scenarios that require context-sensitive analysis and an understanding of sequential or structured data.
> According to Table 1, TexFilter has shown competitive performance in large datasets (e.g., Traffic and Electricity)
> because larger datasets require more contextual structures
> due to their complex relationships.
> However, for the smaller datasets (e.g., ETT and Exchange), since the relationships are relatively simpler,
> using a complex design carries a risk of overfitting.
> For example, in **Figure 10**, the performance with channel-shared filters achieved better results than channel-unique filters,
> which indicates that a simpler architecture is more suitable for smaller dataset.
> In such cases, TexFilter might become overly specialized to the limited data, potentially reducing its generalizability.
>
> We have analyzed TexFilter in **Appendix B** and have also given explanations about the results of Table 1 in **lines 219-224**.
>
> **Q1. Can the authors elaborate on the connection and difference of the two filters?**
>
> A4. Please refer to A1.
>
> **Q2. Can the authors give more details about proposed filters?**
>
> A5.Both PaiFilter and TexFilter belong to frequency filters, and the difference is the filter function $\mathcal{H}\_{filter}$.
> For PaiFilter, $\mathcal{H}\_{filter}$ is equal to $\mathcal{H}\_{\phi}$ where $\mathcal{H}_{\phi}$ is a random initialized learnable weight.
> For TexFilter, $\mathcal{H}\_{filter}$ is equal to $\mathcal{H}\_{\varphi}(\mathcal{F}(\mathbf{Z}))$ where $\mathcal{H}\_{\varphi}()$ is a data-dependent frequency filter learned from the input data.
> We have detailed these in **Equations (1), (4), (5), and (6)**.
>
> **Q3. Can the authors explain the lower performance of TexFilter in some cases?**
>
> A6. Please refer to A3.

---

### Official Review · Reviewer_UnVh · 2024-07-08

**Soundness:** 4
**Presentation:** 4
**Contribution:** 4
**Rating:** 7
**Confidence:** 4

**Summary:**

This paper explores a novel direction that leverages frequency filters directly for time series forecasting, and proposes a simple yet effective framework, namely FilterNet, from a signal processing perspective. It designs two types of learnable frequency filters corresponding to the linear and attention mapping widely used in time series modeling, i.e., PaiFilter and TexFilter. Abundant analysis experiments verify the competitive modeling capability of frequency filters, and extensive experiments on eight real-world datasets demonstrate FilterNet achieves good performances in both accuracy and efficiency over state-of-the-art methods.

**Strengths:**

1.	This paper is well-written overall, and the contribution and novelty of the proposed work are clearly presented.
2.	This paper introduces a novel and straightforward framework based on the frequency filters for time series forecasting. It offers a new direction by incorporating more signal processing theory into time series analysis.
3.	The frequency filter has a simple architecture but demonstrates strong temporal modeling capabilities for time series. Results in Tables 1, 4, and 5 show that the two types of proposed frequency filters are suitable for different datasets and settings, which provides valuable insights for future model design.
4.	This paper seems to have solid and extensive experiments. The experiments are conducted on eight real-world different datasets, and abundant evaluation analysis and visualization experiments are presented.

**Weaknesses:**

1.	The visualization setting for Figure 1 appears to be missing in Appendix C.4, and I recommend adding some references about the spectral bias of neural networks, such as [1].
2.	Some typo errors. In line 282, the complexity of FilterNet should be O(LogL), not O(L LogL).

**Questions:**

1.	In ICML 2024, some works also adopt simple architectures for time series forecasting, such as SparseTSF [2]. Could you further compare the performance of FilterNet with SparseTSF?
2.	The recent literature has seen papers based on foundation models for time series forecasting. Conversely, some works are designed with simple architectures, yet achieve better performance than foundation models. Why can simple models (e.g., FilterNet) also achieve competitive performance?
[1] On the Spectral Bias of Neural Networks, in ICML 2019
[2] SparseTSF: Modeling Long-term Time Series Forecasting with 1k Parameters, in ICML 2024

---

> ### Author Rebuttal · Authors · 2024-08-07
>
> Many thanks for your positive comments and constructive suggestions. We provide a point-by-point response to your comments as follows, and we hope that we address all your concerns below:
>
> **W1. The visualization setting for Figure 1 appears to be missing in Appendix C.4, and I recommend adding some references about the spectral bias of neural networks, such as [1].**
>
> A1. Thanks for your very careful reading. For Figure 1, we generate a simple signal composed of three sine functions, each with a different frequency: low-frequency, middle-frequency, and high-frequency.
> Then, we conducted experiments using iTransformer and FilterNet on the same signal, reported the MSE results, and visualized the prediction values and ground-truth values.
> We will add these in the final version, and add some references about the spectral bias of neural network.
>
> **W2. Some typo errors**
>
> A2. Thanks, we will correct it in the final version.
>
> **Q1. In ICML 2024, some works also adopt simple architectures for time series forecasting, such as SparseTSF [2]. Could you further compare the performance of FilterNet with SparseTSF?**
>
> A3. To further compare the performance of SparseTSF with FilterNet, we conduct experiments on Electricity and Traffic datasets with the look-back window length of 96.
> Since SparseTSF only reported the MSE results, we also report the results of MSE, and the results are shown as follows:
>
> |  Model | | TexFilter| PaiFilter| SparseTSF|
> |:--------:|:----|:----|:----|:----|
> |  Metric | | MSE|MSE|MSE|
> |  Electricity | 96| 0.147| 0.176| 0.209|
> |   | 192| 0.160 |0.185 |0.202 |
> |   | 336| 0.173 |0.202 |0.217 |
> |   | 720| 0.210 |0.242 |0.259 |
> |  Traffic | 96| 0.430| 0.506| 0.672|
> |   | 192| 0.452| 0.508| 0.608 |
> |   | 336| 0.470| 0.518| 0.609|
> |   | 720| 0.498| 0.553| 0.650|
>
> Overall, FilterNet outperforms SparseTSF. The performance of SparseTSF can be validated from its paper (**see Table 12 in SparseTSF**).
> SparseTSF has a simple architecture, which limits its capability for modeling complex relationships.
> This results in SparseTSF achieving worse performance on larger datasets.
> This limitation is also validated in FilterNet's performance comparison.
> On the larger datesets, TexFilter has better performance than PaiFilter, while on smaller datasets, PaiFilter has better than TexFilter.
>
> **Q2. The recent literature has seen papers based on foundation models for time series forecasting. Conversely, some works are designed with simple architectures, yet achieve better performance than foundation models. Why can simple models (e.g., FilterNet) also achieve competitive performance?**
>
> A4. It is a good question. As claimed in SparseTSF, the basis for accurate long-term time series forecasting
> lies in the inherent periodicity and trend of the data.
> Since the periodicity and trend information are very simple, and they can be modeled by a simple network, such as 1k parameters in SparseTSF.
> In a recent paper [1], it discovers that LLM-based methods perform the same or worse than basic ablations,
> yet require orders of magnitude more compute. It shows that replacing language models with simple attention layers,
> basic transformer blocks, randomly-initialized language models, and even removing the language
> model entirely, yields comparable or better performance. Also, in the paper [1],
> it finds that simple methods (patching with one-layer attention) can work so well.
> Combining the findings of FilterNet, we can conclude that simple models can also achieve competitive performance.
>
> [1] Are Language Models Actually Useful for Time Series Forecasting? https://arxiv.org/abs/2406.16964

---

> > ### Comment · Reviewer_UnVh · 2024-08-13
> > **Thank the authors for your rebuttal**
> >
> > Thank the authors for your rebuttal. I have reviewed all comments and the authors’ responses, which effectively addressed my concerns through clear explanations and additional experiments. I recommend incorporating these details into the manuscript or supplementary materials. I have also read the comments from other reviewers but I don't think their reviews can be too convincing to me to re-evaluate this paper. So I keep my original score and overall support the acceptance of this paper.

---

> > > ### Author Response · Authors · 2024-08-13
> > >
> > > Dear Reviewer UnVh,
> > >
> > > we greatly appreciate your continued valuable feedback. Thanks again for  your  constructive suggestions.
> > >
> > > Authors

---

### Official Review · Reviewer_2waa · 2024-07-11

**Soundness:** 2
**Presentation:** 3
**Contribution:** 2
**Rating:** 3
**Confidence:** 4

**Summary:**

The paper proposes to use Fourier transforms with parametrized filters
as layers for time series forecasting models. Filters can either be
learnable parameters directly (called "plain shaping filter") or be
a function in the inputs, gates (called "contextual shaping filter").
In experiments the authors show that their method outperforms
many baselines from the literature.

**Strengths:**

s1. two simple, but promising ideas: i) use Fourier transforms to
  represent time series and ii) represent the filters by gates.
s2. well written paper
s3. standard experiment with improvements over several baselines

**Weaknesses:**

w1. Empirical comparison with results from a closely related baseline, FiLM
  are not shown, and FiLM reports better results for all but one dataset.
- FiLM reports better results for ETTm2, Exchange, and Traffic. The proposed
  FilterNet wins for Weather. Why do you not report those results?
  If FiLM overall is the better performing model, why is FilterNet still
  interesting? How is it simpler than FiLM? Is it maybe faster to train?

w2. Conceptual differences to existing layers using Fourier transforms
  are not discussed in detail.
- So far, we just learn that other architectures also use "other network architectures,
  such as Transformers". But a more detailed delineation would be useful,
  esp. as FiLM is not using Transformers.

minor comments:
- reference [30], FiLM, is missing its venue, NeurIPS.

**Questions:**

Q1. Why do you not report the better results of FiLM?
Q2. If FiLM overall is the better performing model, why is FilterNet still
  interesting? How is it simpler than FiLM?
Q3. Is it maybe faster to train?

**Limitations:**

yes

---

> ### Author Rebuttal · Authors · 2024-08-07
>
> We appreciate your review regarding our paper. We would like to respond to your comments and hope that we address all your concerns below:
>
> **W1. Empirical comparison with results from a closely related baseline, FiLM are not shown, and FiLM reports better results for all but one dataset.**
>
> A1.
> 1. We have compared FilterNet with the newest representative SOTA methods, i.e., the Transformer-based method iTransformer (ICLR, 2024) and the frequency-based method FITS (ICLR, 2024).
> **Note that FITS has already compared with FiLM (NeurIPS, 2022). According to the reported results from FITS (Tables 8 and 9), it shows that FITS outperforms FiLM**. Therefore, we only compare with FITS and do not choose FiLM as a baseline.
> 2. **Note that FiLM conducted experiments with various input length (as described in the caption of Table 1 in FiLM), and the
> input length is tuned for best forecasting performance**. In contrast, in the FilterNet, we follow the settings of iTransformer (ICLR, 2024), PatchTST (ICLR, 2023), and FEDformer (ICML, 2022), and set the input length fixed, such as 96 in Table 1. As a result, direct comparison to the findings in the original FiLM paper is not feasible.
>    To further compare FilterNet with FiLM, we conduct additional experiments with a fixed input length of 96, and the results are shown in below:
>
> |  Model | | TexFilter| | PaiFilter| | FiLM| |
> |:--------:|:----|:----|:----|:----|:----|:----|:----|
> |  Metric | | MSE|MAE | MSE|MAE | MSE|MAE | MSE|MAE  |
> |ETTm2|96	|0.180|	0.262|	0.174|	0.257|	0.183|	0.266|
> | |192|	0.246|	0.306|	0.240|	0.301|	0.248|	0.305|
> | |336|	0.312|	0.348|	0.297|	0.339|	0.309|	0.343|
> | |720|	0.414|	0.405|	0.397|	0.398|	0.409|	0.400 |
> |Electricity|96| 	0.147| 	0.245| 	0.176| 	0.264|	0.198| 	0.274|
> | | 192| 	0.160| 	0.250| 	0.185| 	0.270| 	0.198| 	0.278|
> | | 336| 	0.173| 	0.267| 	0.202| 	0.286| 	0.217| 	0.300|
> | | 720| 	0.210| 	0.309| 	0.242| 	0.319| 	0.278| 	0.356|
> |Exchange|96|	0.091|	0.211|	0.083|	0.202|	0.087|	0.211|
> | |192|	0.186|	0.305|	0.174|	0.296|	0.182|	0.308|
> | |336|	0.380|	0.449|	0.326|	0.413|	0.351|	0.427|
> | |720|	0.896|	0.712|	0.840|	0.670|	0.906|	0.715|
> |Traffic|96|	0.430|	0.294|	0.506|	0.336|	0.646|	0.384|
> | |192|	0.452|	0.307|	0.508|	0.333|	0.600|	0.361|
> | |336|	0.470|	0.316|	0.518|	0.335|	0.609|	0.367|
> | |720|	0.498|	0.323|	0.553|	0.354|	0.691|	0.425|
> |Weather|96|	0.162|	0.207|	0.164|	0.210|	0.196|	0.237|
> | |192|	0.210|	0.250|	0.216|	0.252|	0.239|	0.271|
> | |336|	0.265|	0.290|	0.273|	0.294|	0.289|	0.306|
> | |720|	0.342|	0.340|	0.344|	0.342|	0.360|	0.351|
>
> Also, these results of FiLM with the input length of 96 can be validated by other papers, such as [1] (see its Table 13).
> According to the above table, we can find that overall, FilterNet outperforms FiLM. We will add these results in the final version.
>
> [1]TIMEMIXER: DECOMPOSABLE MULTISCALE MIXING FOR TIME SERIES FORECASTING, ICLR, 2024
>
> **W2. Conceptual differences to existing layers using Fourier transforms are not discussed in detail.So far, we just learn that other architectures also use "other network architectures, such as Transformers". But a more detailed delineation would be useful, esp. as FiLM is not using Transformers.**
>
> A2.
> We organize the time series modeling using Fourier transforms from two perspectives: the incorporated neural network and the filter types.
> Autoformer and FEDformer build on the Transformer architecture and enhance its effectiveness by leveraging Fourier technologies.
> FreTS builds on MLP and enhance its modeling capability by leveraging frequency technologies.
> Although FITS and FiLM do not incorporate network architectures, they function as low-pass filters.
> In contrast, FilterNet is an all-pass filter to avoid information loss
> and also does not incorporate network architectures, ensuring high efficiency.
>
> We will reorganize the discussion in the final version to make it clearer.
>
> **minor comments**
>
> A3. Thanks for your comments. We will correct it in the final version.
>
> **Q1. Why do you not report the better results of FiLM?**
>
> A4. Please refer to A1.
>
> **Q2. If FiLM overall is the better performing model, why is FilterNet still interesting? How is it simpler than FiLM?**
>
> A5.
> 1. About the performance, please refer to A1.
> 2. About the efficiency, please refer to A6.
> 3. About the architecture, FiLM first leverages a Legendre Projection Layer to obtain a compact representation and then employs a Frequency Enhance Layer to learn the projection of representations.
>    In contrast, FilterNet uses only a Frequency Filter Block to model the temporal dependencies.
>
> **Q3. Is it maybe faster to train?**
>
> A6. Both FiLM and FilterNet conduct complex multiplication between the input and the learnable weights.
> However, the learnable weights in FiLM are 3-dimensional $W \in \mathbf{C}^{L\times N\times N}$ where N is the number of Legendre Polynomials,
> while in FilterNet they are 1-dimensional $W \in \mathbf{C}^{1\times L}$.
> This makes the calculation in FiLM more complicated than in FilterNet.
> To further compare the efficiency of FiLM with FilterNet, we conduct efficiency experiments on the Exchange dataset, and the results are shown below:
>
> |Model | Memory Usage (G)|  Training Time (s/epoch)|
> |:--------:|:----|:----|
> |  FiLM | 0.9 |  48.2 |
> |  FilterNet | 0.5 |  1.5 |
>
> From the table, we can find that FilterNet has faster training speed and less memory usage than FiLM.
> The efficiency of other models is shown in Figure 8.

---

> ### Author Response · Authors · 2024-08-12
>
> Dear Reviewer 2waa,
>
> We appreciate your time and effort in reviewing our paper. As the discussion period is drawing to a close, we wanted to kindly  inquire if there are any further concerns.
> Your feedbacks are really important to us, and we are also looking forward to further discussions with you before the discussion phase ends.
> We value your input and would appreciate the opportunity to respond.
>
> Thank you again for your valuable time and effort.
>
> Best,
>
> Authors

---

> > ### Author Response · Authors · 2024-08-13
> >
> > Dear Reviewer 2waa,
> >
> > With the discussion period nearing its end and less than 18 hours remaining,
> > we kindly urge you to respond to our rebuttal.

---

### Official Review · Reviewer_ufqs · 2024-07-30

**Soundness:** 3
**Presentation:** 3
**Contribution:** 2
**Rating:** 4
**Confidence:** 4

**Summary:**

This paper investigates an interesting direction from a signal processing perspective, making a new attempt to apply frequency filters directly for time series forecasting. A simple yet effective architecture called FilterNet is proposed, utilizing two types of frequency filters to achieve forecasting. Comprehensive empirical experiments on eight benchmarks confirm the superiority of this paper's proposed method in terms of effectiveness and efficiency.

**Strengths:**

1. The paper is in good writing shape and easy to follow.
2. The proposed method seems technically solid.
3. Extensive experiments on many datasets have been conducted to verify the effectiveness of the approach.

**Weaknesses:**

1. The marginal improvement over existing approaches, as shown in Table 1, raises doubts about the compelling rationale for adopting the proposed method.
2. The discussion on related work appears lacking. It is common knowledge that frequency filters are widely used in traditional time series analysis methods and signal processing. Therefore, a detailed exploration of the connections and distinctions between them is warranted in the related work section.
3. The experiments have space to improve. Including more recent baselines for comparison, particularly deep learning methods that involve frequency processing, would add value. Additionally, a thorough investigation into the impact of hyperparameters is missing from the study.

**Questions:**

Please address the questions in the weaknesses.

---

> ### Author Rebuttal · Authors · 2024-08-07
>
> Many thanks for your constructive comments and suggestions. We provide a point-by-point response to your comments below:
>
> **W1. The marginal improvement over existing approaches, as shown in Table 1, raises doubts about the compelling rationale for adopting the proposed method.**
>
> A1. The primary contribution of this paper lies in offering a novel viewpoint from signal processing and devising a simple yet effecitve architecture centered around a Frequency Filter Block.
> In contrast to prior research, FilterNet show remarkable efficiency (**as shown in Figure 8**) while simultaneously achieving state-of-the-art performance.
> Furthermore, we extensively investigate the efficacy and insights of frequency filters via introducing two distinct types of frequency filters, each exhibiting diversified performances across datasets of different scales. This exploration reveal the impact of frequency filters on time series forecasting and indicates that designing tailored architectures adaptive to data scale may enlighten a new path for future research endeavors
>
> **W2. The discussion on related work appears lacking. It is common knowledge that frequency filters are widely used in traditional time series analysis methods and signal processing. Therefore, a detailed exploration of the connections and distinctions between them is warranted in the related work section.**
>
> A2.
> In signal processing, filters are employed to selectively enhance or attenuate specific frequency components of a signal.
> There are various types of filters, including low-pass filter, band-pass filter, Butterworth filter, and Chebyshev filter, etc.
> The choice of filter type is highly dependent on the signal characteristics and the desired outcome.
> However, in time series forecasting scenarios, the signal characteristics and desired outcomes are unknown in advance.
> This uncertainty complicates the selection and design of appropriate filters, necessitating learnable or data-driven approaches to effectively capture and forecast the underlying patterns in the time series data.
> In FilterNet, unlike FITS and FiLM, which directly use low-pass filters, we leverage learnable frequency filters.
> Additionally, we design two filters that are inspired by Transformer and MLP architectures.
> This approach allows the model to learn the most relevant frequency components, improving forecasting performance without requiring prior knowledge of the signal characteristics.
>
> We will add these discussions in the final version.
>
> **W3. The experiments have space to improve. Including more recent baselines for comparison, particularly deep learning methods that involve frequency processing, would add value. Additionally, a thorough investigation into the impact of hyperparameters is missing from the study.**
>
> A3.
> 1. We choose four categories of baselines: Frequency-based, Transformer-based, MLP-based, and TCN-based methods,
>    including the latest SOTA methods, i.e., the Transformer-based iTransformer (ICLR, 2024) and the Frequency-based FITS (ICLR, 2024).
>    We further compare FilterNet with SparseTSF (ICML, 2024) (please refer to A3 in Response to Reviewer UnVh) and FiLM (NeurIPS, 2022) (please refer to A1 in Response to Reviewer 2waa).
>    Additionally, we include Koopa (NeurIPS, 2023), which is designed for addressing non-stationary issues in time series by disentangling time-invariant and time-variant dynamics through Fourier Filter, as shown below:
>
> |  Model | | TexFilter| | PaiFilter| | Koopa| |
> |:--------:|:----|:----|:----|:----|:----|:----|:----|
> |  Metric | | MSE|MAE | MSE|MAE | MSE|MAE |
> |Exchange | 96	| 0.091| 	0.211| 	0.083| 	0.202| 	0.090| 	0.210|
> | | 192	| 0.186	| 0.305| 	0.174| 	0.296| 	0.188| 	0.310|
> | | 336	| 0.380 | 0.449| 	0.326| 	0.413| 	0.371| 	0.444|
> | | 720	| 0.896 | 0.712| 	0.840| 	0.670| 	0.896| 	0.713|
> | Traffic |96|	0.430|	0.294|	0.506|	0.336|	0.517|	0.349|
> | |192|	0.452|	0.307|	0.508|	0.333|	0.529|	0.358|
> | |336|	0.470|	0.316|	0.518|	0.335|	0.540|	0.361|
> | |720|	0.498|	0.323|	0.553|	0.354|	0.591|	0.386|
>
> We choose two datasets: one smaller dataset (Exchange) and one larger dataset (Traffic), and we compare them with the look-back window length of 96.
> From the table, we can find that FilterNet outperforms Koopa whether the dataset is smaller or larger.
>
> 2. The bandwidth of frequency filters holds significant importance in the functionality of filters,
>    and we have analyzed the bandwidth parameter in the experimental part (see **lines 254-266**).
>    For PaiFilter, there are no hyperparameters other than a bandwidth parameter.
>    For TexFilter, there is one hyperparameter $K$, and we conduct parameter sensitivity experiments about K as below:
>
> |   | TexFilter| |
> |:--------:|:----|:----|
> | Metric  |  MSE|MAE |
> |  K=1 | 0.147 | 0.245 |
> |  K=2 | 0.145 | 0.243 |
> |  K=3 | 0.142| 0.242|
>
> Although adding layers can improve performance, considering resource costs with additional layers, we use a one-layer architecture (i.e.,
> K=1), similar to DLinear, FreTS, and FiLM, which also use a one-layer architecture.
> We will add the analysis in the final version.

---

> ### Author Response · Authors · 2024-08-12
>
> Dear Reviewer ufqs,
>
> Thank you very much again for your time and efforts in reviewing our paper. We kindly remind that our discussion period is closing soon. We just wonder whether there is any further concern and hope to have a chance to respond before the discussion phase ends.
>
> Many thanks,
>
> Authors

---

> ### Comment · Reviewer_ufqs · 2024-08-12
> **Reviewer Response**
>
> Thank you for addressing my questions. I still have the following concerns:
> 1. The authors have restated the technical contributions and efficiency achieved by the proposed method; however, the improvement in predictive performance is **neither significant nor consistent across all datasets**, particularly on the Traffic dataset. How many experimental runs were conducted for the reported results? From my understanding, the results of deep learning algorithms can vary across different runs. Can you guarantee that your improvements are statistically significant across all datasets? Additionally, what are the standard deviations across all runs?
> 2. The performance of TexFilter and PaiFilter varies significantly across different datasets. What is the underlying reason for this fluctuation? It’s difficult to determine when to use TexFilter versus PaiFilter. A more rigorous justification or empirical study is needed. Otherwise, I would prefer a model that demonstrates consistent performance across all scenarios, such as PatchTST and [1, 2].
> 3. Although the authors provide several results comparing FITS and iTransformer, more recent and powerful baselines, such as TimeXer [1] and Timer [2], should also be considered.
> 4. Why didn't the paper compare the efficiency of frequency-based methods (e.g., FITS, FilM) with the proposed method? Compared to other types of models, it is more necessary to compare it with them. Moreover, it would be beneficial to include TimeXer and Timer in the efficiency comparison as well.
>
> Given the above concerns, I don't believe this paper is ready to publish on a top-tier venue like NeurIPS.
>
> Reference:
>
> [1] TimeXer: Empowering Transformers for Time Series Forecasting with Exogenous Variables. arXiv 2024.
>
> [2] Timer: Generative Pre-trained Transformers Are Large Time Series Models. ICML 2024.

---

> > ### Author Response · Authors · 2024-08-12
> >
> > Thanks for your feedback. We provide a point-by-point response to your new concerns below.
> >
> > **1. ....however, the improvement in predictive performance is neither significant nor consistent across all datasets, particularly on the Traffic dataset.**
> >
> > A1. We think your comment is not professional about improvements in the domain of time series forecasting. We don't understand what level of improvement is considered sufficient in your opinion.
> > As you mentioned, TimeXer, what you refer to as a more recent and powerful method,
> > its performance is worse than iTransformer on the ECL, Traffic, and Weather datasets, and worse than RLinear on ETTh2 dataset (please see its Table 8). Similarly, iTransformer shows weaker performance on the ETTh and ETTm datasets. Are you also questioning their performance improvements, too? It is **ridiculous** the failure to achieve optimal performance across all datasets could also be a reason of rejection.
> >
> > **2. How many experimental runs were conducted for the reported results? From my understanding, the results of deep learning algorithms can vary across different runs. Can you guarantee that your improvements are statistically significant across all datasets? Additionally, what are the standard deviations across all runs?**
> >
> > A2. Your question seems to be not related to our submission.
> > We are wondering if you generate this review comments by Generative AI (e.g., ChatGPT).
> > If you have run the state-of-the-art baselines in the domain of time series forecasting (e.g., iTransformer, PatchTST),
> > you would know these forecasting methods are usually stable thus they did not report std.
> > We conducted the experiments three times with three seeds, and reported the average performance with the very stable results and less std.
> > Thus we followed previous settings to report the performance.
> > The average improvement of FilterNet over all baseline models is statistically significant at the confidence of 95% (**please see in line 218 of our paper**).
> >
> > **3. The performance of TexFilter and PaiFilter varies significantly across different datasets. What is the underlying reason for this fluctuation? It’s difficult to determine when to use TexFilter versus PaiFilter. A more rigorous justification or empirical study is needed. Otherwise, I would prefer a model that demonstrates consistent performance across all scenarios, such as PatchTST and [1, 2].**
> >
> > A3. In the literature, compared with other model architectures, MLP-based and Transformer-based methods have achieved competitive performance. Inspired by the two paradigms, we design two corresponding types of filters. i.e., PaiFilter and TexFilter. PaiFilter offer stability and efficiency, making them suitable for tasks with static relationships. In contrast, TexFilter provides the flexibility to capture dynamic dependencies, excelling in tasks that require context-sensitive analysis. As shown in Table 1, PaiFilter performs well on small datasets (e.g., ETTh1), while TexFilter excels on large datasets (e.g., Traffic) due to the ability to model the more complex and contextual correlations present in larger datasets. This dual approach allows FilterNet to effectively handle a wide range of data types and forecasting scenarios, combining the best aspects of both paradigms to achieve superior performance.
> >
> > **We have analyzed the two filters in Appendix B and have also given explanations about the results of Table 1 in lines 219-224**.
> >
> > You again mentioned a **ridiculous** point. We don't know why the proposed method should be one model. Like DLinear, it also has studied three different variants, i.e., NLinear, Linear, and DLinear. Can you say you do not want to see the Linear family models in one paper?
> >
> > **4. ... comparing FITS and iTransformer, more recent and powerful baselines, such as TimeXer [1] and Timer [2], should also be considered.**
> >
> > A4.
> > We have also compared FilterNet with SparseTSF (ICML, 2024) and Koopa (NeurIPS, 2023), and we are very confused that why we should compare with TimeXer which as only one preprint paper, and Timer as a generative pre-trained large model.
> > **What is even more crazy** is that you recommend we include Timer in the efficiency comparison. Timer is a pre-trained model, which has a 28B pre-trained parameters (see its Table 9). It is brutal.
> >
> > **5. Why didn't the paper compare the efficiency of frequency-based methods (e.g., FITS, FilM) with the proposed method? Compared to other types of models, it is more necessary to compare it with them.Moreover, it would be beneficial to include TimeXer and Timer in the efficiency comparison as well.**
> >
> > A5. We have compared the efficiency with FiLM, please refer to A6 in Response to Reviewer 2waa.
> > For FITS, it is a lightweight method and has 10k parameters. Obviously, FITS is more efficient. However, our model is not designed for a lightweight model, but a model that balances efficiency and effectiveness.
> > As for the efficiency comparison with TimeXer and Timer, please refer A4.

---

> > > ### Comment · Reviewer_ufqs · 2024-08-13
> > > **Reviewer response**
> > >
> > > Thank you for the rebuttal. Here is my feedback:
> > >
> > > 1. Sure, the answer is not generated by GPT. The reviewer has worked in the field of time series analysis for many years since statistical methods. From a statistical standpoint, it is crucial or even compulsory to conduct rigorous statistical tests to evaluate improvements, especially when the improvements are insignificant. The reason you mentioned (other papers not reported) is not convincing and the evaluation goes against scientific rigor.
> > >
> > > 2. For instance, DLinear offers three variants, so why did the researchers only compare one? In future comparisons with your model, which variant should be selected? Which variant do you recommend? This is a very important problem in practice, especially for the industry.
> > >
> > > 3. The reason for not including TimeXer and Timer is reasonable.
> > >
> > > 4. My evaluation is centered on the technical novelty/contributions and practical results achieved by the proposed method. While the reviewer isn't particularly impressed by the technical novelty/contributions, the reviewer prioritizes methods that demonstrate strong performance in real-world applications.
> > >
> > > I have adjusted my ratings and would like to leave the discussion to ACs and other reviewers.

---

> > > > ### Author Response · Authors · 2024-08-13
> > > >
> > > > Thanks for your feedback. We provide a response to your main concerns below.
> > > >
> > > > 1. We will consider including the standard deviations in the final version, although none of our baselines—such as iTransformer, PatchTST, FEDformer, TimesNet, DLinear, RLinear, FITS, FreTS, MICN, Pyraformer, Autoformer, and Informer—report standard deviations in their papers.
> > > >
> > > > 2. In the literature, it is quite common to design different model variants to adapt to various datasets.
> > > >    For instance, FEDformer introduced two model variants, FEDformer-f and FEDformer-w;
> > > >    PatchTST designed two model variants, PatchTST/64 and PatchTST/42;
> > > >    DLinear included three model variants, DLinear, NLinear, and Linear.
> > > >   Switching between these variants typically involves adjusting a configuration parameter,
> > > >    such as the "version" parameter for FEDformer, "patch_len" for PatchTST, or "model" for DLinear.
> > > > Typically, a model name is used generically to refer to all variants, with the specific variant being distinguished by its corresponding configuration parameters.
> > > >
> > > >    For our proposed FilterNet, as mentioned in the previous rebuttal, TexFilter has demonstrated competitive performance on larger datasets (e.g., Traffic and Electricity) due to the need for more contextual structures, while PaiFilter performs well on smaller datasets (e.g., ETTh1).
> > > >
> > > >     In practice, especially for the industry, the balance between efficiency and effectiveness is what matters most. And which model is more suitable requires actual comparison and testing.
> > > >
> > > > 3. As highlighted in our previous rebuttal, the primary contribution of this paper lies in offering a novel viewpoint from signal processing and devising a simple yet effective architecture centered around a Frequency Filter Block.
> > > >   FilterNet shows remarkable efficiency while simultaneously achieving state-of-the-art performance. Furthermore, we extensively investigate the efficacy and insights of frequency filters via introducing two distinct types of frequency filters, each exhibiting diversified performances across datasets of different scales. This exploration reveal the impact of frequency filters on time series forecasting and indicates that designing tailored architectures adaptive to data scale may enlighten a new path for future research endeavors.

---

### Decision · Program_Chairs · 2024-09-25

**Decision:**

Accept (poster)

**Comment:**

This paper proposes a novel architecture called FilterNet for time series forecasting, which aims to balance efficiency and effectiveness by introducing frequency filters.
The contributions were generally well-received, with many reviewers acknowledging its technical soundness.
However, some reviewers also raised concerns that the technical contributions might be slightly below the standards expected at NeurIPS.
The authors' rebuttal effectively addressed many of the reviewers' concerns.
Nevertheless, when considering the technical standards and the relative evaluation of other submissions, the paper's priority is lower, which recommends the final decision to reject the submission.